# ConceptFlow: Unified Framework for Personalized Image Generation

## Abstract

Personalized image generation is an appealing area of research within controllable image generation due to its diverse potential applications. Despite notable advancements, generating images based on single or multiple concepts remains challenging. For single-concept generation, it is difficult to strike a balance between identity preservation and prompt alignment, especially in complex prompts. When it comes to multiple concepts, creating images from a single prompt without extra conditions, such as layout boxes or semantic masks, is problematic due to significantly identity loss and concept omission. In this paper, we introduce ConceptFlow, a comprehensive framework designed to tackle these challenges. Specifically, we propose ConceptFlow-S and ConceptFlow-M for single-concept generation and multiple-concept generation, respectively. ConceptFlow-S introduces a KronA-WED adapter, which integrates a Kronecker adapter with weight and embedding decomposition, and employs a disentangled learning approach with a novel attention regularization objective to enhance single-concept generation. On the other hand, ConceptFlow-M leverages models learned from ConceptFlow-S to directly generate multi-concept images without needed of additional conditions, proposing Subject-Adaptive Matching Attention (SAMA) module and layout consistency guidance strategy. Our extensive experiments and user study show that ConceptFlow effectively addresses the aforementioned issues, enabling its application in various real-world scenarios such as advertising and garment try-on.

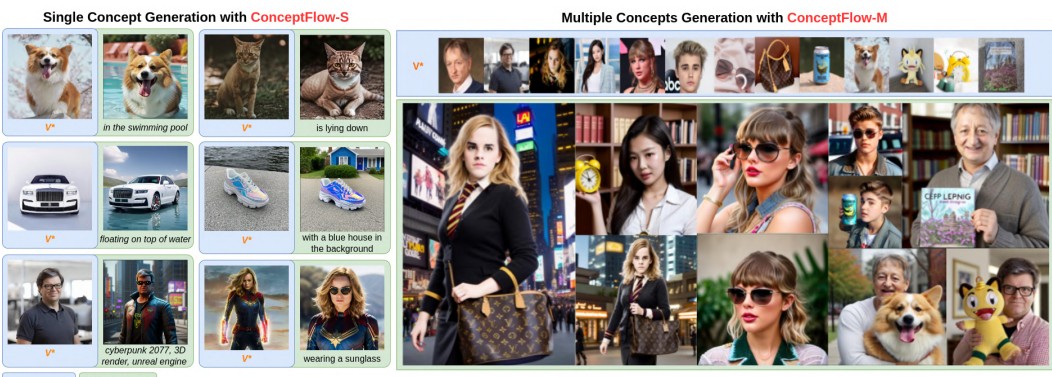

Figure 1: The generated images of our proposed ConceptFlow framework for both single concept and multiple-concept scenarios.

## 1 Introduction

Personalized image generation aims to generate customized images of the given concepts (or subjects) based on the text descriptions, e.g. a selfie image of a celebrity with your pet. This field has attracted an increasing number of attention due to its potential in various applications, such as story telling, advertisement and garment try-on. The core tasks in personalized image generation are generating images from individual concepts and integrating multiple concepts into images. Despite considerable advancements, each task presents its own challenges.

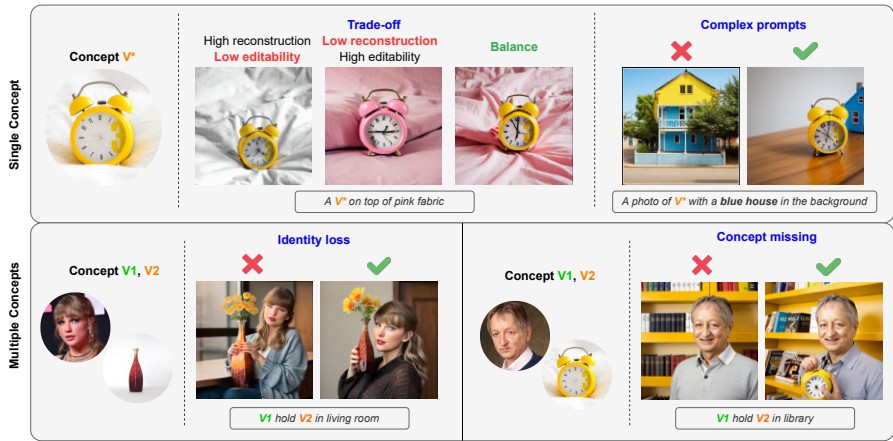

Figure 2: Challenges in personalized image generation. Top: Trade-off between reconstruction and editability, especially in complex prompts in single concept generation. Bottom: Identity loss and concept missing issues in generating multiple concepts.

Although significant progress has been made (Gal et al., 2023; Ruiz et al., 2023; Kumari et al., 2023; Chen et al., 2024; Gu et al., 2024), the trade-off between reconstruction (i.e., identity preservation) and editability (i.e., prompt alignment) remains a challenge in single concept generation Parameter-efficient fine-tuning (PEFT) methods (Ryu; Gu et al., 2024) using low-rank adaptation (LoRA) (Hu et al., 2024) techniques to enhance efficiency, but they often suffer from the low-rank assumption, which hampers their ability to preserve fine-grained identity details. DisenBooth (Chen et al., 2024) introduces disentangled learning to significantly improve prompt alignment. However, generating images that accurately reflect intricate prompts, such as the interaction between desired concepts and other subjects in prompts, continues to be an obstacle. As illustrated in Figure 2, the generated images lack the concept of the *yellow clock*, and its color is incorrectly associated with the *blue house*. This suggests that the attention maps for the concept tokens are not effectively focused on the appropriate areas in the image.

For multi-concept generation, directly creating images containing multiple concepts from separately learned models using only a text prompt remains challenging due to significant identity loss and concept omission problems (see Figure 2). These issues arise because concepts are typically learned in contexts where they are the main subjects at the center of the layout. However, in a composite multi-concept image, their scale, position, and pose can vary significantly. To overcome these limitations, existing methods (Gu et al., 2024; Kong et al., 2024) rely on the additional conditions such as bounding boxes, segmentation masks, and the sampling process is performed with different prompts for different regions in the image. However, relying on additional conditions sometimes produces unsatisfied results, particularly in scenarios requiring interaction where concepts occlude each other. Moreover, creating these conditions is not easy for ordinary users, which restricts their capacity for practical applications.

To this end, we propose ConceptFlow-S and ConceptFlow-M to address the above mention issues. For single concept learning and generation, ConceptFlow-S introduces a module called **Kron**ecker **A**daptation with **W**eight and **E**mbedding **D**ecomposition (KronA-WED) to enhance the reconstruction capability by offering high rank updated matrices while keeping the model size. We also employ disentangled learning (Chen et al., 2024) along with a novel attention regularization objective during fine-tuning to refine the cross-attention map of concept tokens, thereby further enhancing the editability. As a result, ConceptFlow-S balances the trade-off effectively. For multi-concept generation, ConceptFlow-M fuses individually learned models from ConceptFlow-S by gradient fusion algorithm (Gu et al., 2024), thereby introducing **S**ubject-**A**daptive **M**atching **A**ttention (SAMA) module to preserve the appearance of multiple concepts, and layout consistency guidance strategy to address the issue of concepts missing. The illustrations for generated images using our ConceptFlow framework are shown in Figure 1. We conduct extensive experiments and user study to demonstrate the performance of ConceptFlow over the state-of-the-art methods in single and multiple concepts generation. The ablation studies validate our design choices and highlight the effectiveness of each component.

In summary, our contributions are as follows:

- We introduce the ConceptFlow framework, including the ConceptFlow-S component for robust single concept learning and generation, and the ConceptFlow-M component to create images of multiple concepts without the need of spatial guidance.
- ConceptFlow-S introduces KronA-WED adapter along with a disentangled learning and a novel attention regularization objective to balance the trade-off between reconstruction and editability.
- ConceptFlow-M proposes a multi-concept appearance preservation strategy with SAMA module and layout consistency guidance to address the issue of missing concepts.
- We showcase the effectiveness of ConceptFlow through extensive experiments and user study.

## 2 RELATED WORK

### 2.1 SINGLE-CONCEPT GENERATION

Despite considerable advancements, balancing reconstruction and editability continues to pose a challenge for optimization-based methods (Gal et al., 2023; Voynov et al., 2023; Ruiz et al., 2023; Kumari et al., 2023). Parameter-efficient fine-tuning methods based on LoRA (Hu et al., 2024) techniques (Ryu; Gu et al., 2024) significantly reduce the fine-tuning cost. Still, they often encounter difficulties in capturing complex details of concepts due to the low-rank assumption. To overcome this issue, LyCORIS (Yeh et al., 2023) adopted Kroncker Adapter (KronA) (Edalati et al., 2022) for fine-tuning diffusion models. However, only integrating KronA with purely joint weight-embedding tuning methods risks further compromising editability. DisenBooth (Chen et al., 2024) proposed a disentangled learning strategy to improve the editability of joint embedding-weight tuning methods. However, attaining effective alignment with complex prompts remains challenging due to wrongly activated regions in token attention maps.

In merging separately learned concepts via gradient fusion (Gu et al., 2024) for multiple concepts generation, Embedding-Decomposed Low-Rank Adaptation (ED-LoRA) (Gu et al., 2024) is the most commonly used adapter. Taking it as the baseline, our focus is on improving both the editability and reconstruction capabilities, consequently facilitating the multi-concept generation process.

### 2.2 MULTIPLE-CONCEPT GENERATION

Recent advancements have pushed the task of customization further by attempting to inject multiple novel concepts into a model simultaneously. Break-a-Scene (Avrahami et al., 2023) and SVD-iff (Han et al., 2023) learn individual concepts within images containing multiple concepts. However, these methods required access to ground truth training data, limiting their flexibility in practical applications. Custom Diffusion (Kumari et al., 2023) achieved this through a joint optimization loss for all concepts. Mix-of-Show (MoS) (Gu et al., 2024) introduced gradient fusion to allow the merging of multiple separately fine-tuned models. However, generating images containing multiple concepts using sorely prompts with the fused model remains challenging due to the identity loss of the mentioned concepts and the issue of missing concepts. MoS proposed region sampling to overcome these problems. Still, this method requires extra conditions like bounding boxes, human poses, or sketches, and it is not effective when concepts interact or occlude with each other. OMG (Kong et al., 2024) addresses occlusions by first generating a layout image, then blending noise into specific regions using predicted masks. Nonetheless, its main limitation is the mismatch between predicted masks and the actual shapes of the subjects, causing identity loss and inaccurate blending.

## 3 PRELIMINARIES

We provide an overview of the foundational preliminaries in Appendix A.1, which are fine-tuning Stable Diffusion model for personalization and weight decomposition (DORA) Liu et al. (2024).

**Parameter-efficient Fine-tuning for Personalization.** Based on the assumption that updates made during the fine-tuning exhibit a low intrinsic rank, Low-Rank Adaptation (LoRA) (Hu et al., 2024) significantly reduces the training parameters by modeling the weight update $\Delta W \in \mathbb{R}^{d \times k}$ using a low-rank decomposition, expressed as $BA$, where $B \in \mathbb{R}^{d \times r}$ and $A \in \mathbb{R}^{r \times k}$, with $r \ll \min(d, k)$.

Then, the fine-tuned weight $W'$ can be formulated as:

$$W' = W_0 + \underline{\Delta W} = W_0 + \underline{BA}, \tag{1}$$

where $W_0 \in \mathbb{R}^{d \times k}$ is the pre-trained weight matrix and the underlined parameters are being trained during the fine-tuning process. Kronecker product decomposition is an factorization method based on Kronecker product ($\otimes$) and it does not depend on the low-rank assumption. Edalati et al. (2022) replaces low-rank decomposition in LoRA with Kronecker decomposition to develop the Kronecker Adapter (KronA). Specifically, the updated weight $\Delta W \in \mathbb{R}^{d \times k}$ is expressed as $A \otimes B$, where $A \in \mathbb{R}^{a_1 \times a_2}$ and $B \in \mathbb{R}^{b_1 \times b_2}$, with $a_1 \times b_1 = d$ and $a_2 \times b_2 = k$. More details about LoRA (Hu et al., 2024) and KronA (Edalati et al., 2022) are provided in Appendix A.1

**Disentangled learning.** Chen et al. (2024) proposed learning separate textual and visual embeddings to prevent the mixing of a concept's identity with irrelevant details during fine-tuning, thereby significantly enhancing the editability capability. Specifically, besides the text features $f_s$ from the given prompt, they extracted image features $f_i$ from training images and combine $f_s$ with $f_i$ for the fine-tuning process, which had three learning objectives: denoising loss $L_{denoise}$, weak denoising loss $L_{w-denoise}$, and contrastive embedding loss $L_{con}$. Please see Appendix A.1.5 for more details.

**Appearance Matching Self-Attention (AMA)** is introduced by Nam et al. (2024) to enhance the expressiveness of the appearance of a single concept by injecting a real reference image into the target denoising process. This technique includes an additional reference branch where the DDIM inverted latent representation (Couairon et al., 2023) of the given reference image accompanies the main denoising target branch. At timestep $t$, it preserves the target structure by replacing only the target appearance path $\mathbf{V}_t^{trg}$ with warped reference values $\mathbf{V}_t^{ref \to trg}$, which can be formulated as:

$$\mathbf{V}_t^{ref \to trg} = \mathcal{W}(\mathbf{V}_t^{ref}; \mathbf{F}_t^{ref \to trg}), \tag{2}$$

where $\mathcal{W}$ represents the warping operation (Truong et al., 2021) and $\mathbf{V}_t^{ref}$ is the projected value in the self-attention module from reference branch. The dense displacement field from the reference to the target, denoted as $\mathbf{F}_t^{ref \to trg} \in \mathbb{R}^{H \times W \times 2}$, is derived using the argmax operation on the matching cost $\mathbf{C}_t \in \mathbb{R}^{(H \times W) \times (H \times W)}$, which are pairwise cosine similarities between the feature descriptors $\psi_{t-1}^{trg} \in \mathbb{R}^{H \times W \times D}$ and $\psi_{t-1}^{ref} \in \mathbb{R}^{H \times W \times D}$ from both the reference and target branches:

$$\mathbf{C}_t(x, y) = \frac{\psi_{t-1}^{trg}(x) \cdot \psi_{t-1}^{ref}(y)}{\|\psi_{t-1}^{trg}(x)\| \|\psi_{t-1}^{ref}(y)\|}, \tag{3}$$

where $x, y \in [0, H) \times [0, W)$, $\| \cdot \|$ denotes $l_2$ normalization. In the later sections, we omit the subscripts about timesteps from these notations for simplicity.

## 4 PROPOSED METHOD: CONCEPTFLOW

### 4.1 CONCEPTFLOW-S FOR SINGLE CONCEPT LEARNING AND GENERATION

Taking ED-LoRA (Gu et al., 2024) adapter as the baseline, we introduce a new adapter called **Kron**ecker **A**daptation with **W**eight and **E**mbedding **D**ecomposition (KronA-WED) and a fine-tuning strategy that combines disentangled learning (Chen et al., 2024) with a novel attention regularization objective to achieving the balance of reconstruction and editability. The pipeline of ConceptFlow-S for single concept learning is depicted in Figure 3.

**KronA-WED Adapter.** The low-rank assumption can impede the ability of LoRA-based methods (Ryu; Gu et al., 2024) in capturing complex patterns of concepts in single-concept learning (see Figure 4a). Increase the rank value can enhance this capability but it also results in larger model sizes. To relax the low-rank assumption on updated weights and keep the model size small, we propose a more flexible approach by extending the ED-LoRA architecture (Gu et al., 2024) with Kronecker Adapter (KronA) (Edalati et al., 2022). We also perform the weight decomposition technique (Liu et al., 2024) on $W'$ to enhance the learning capability of the adapter, thereby forming a new adapter called KronA-WED. The fine-tuned weights $W'$ are specified as follow:

$$W' = \underline{m} \frac{W_0 + \underline{\Delta W}}{\|W_0 + \underline{\Delta W}\|_c} = \underline{m} \frac{W_0 + \underline{A \otimes B}}{\|W_0 + \underline{A \otimes B}\|_c}, \tag{4}$$

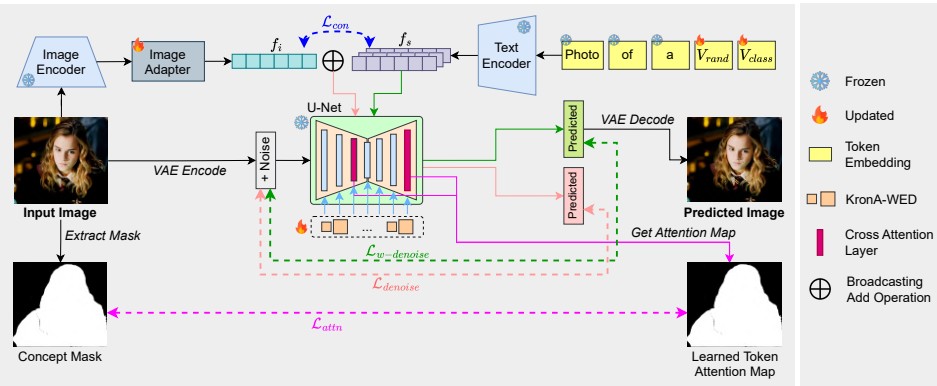

Figure 3: The pipeline of ConceptFlow-S for single concept learning and generation. We fine-tune the newly-added tokens ($V_{rand}$ and $V_{class}$), our proposed KronA-WED adapters, and image adapter with a novel attention regularization objective $\mathcal{L}_{attn}$. We also employ disentangled learning (Chen et al., 2024), including $\mathcal{L}_{denoise}$, $\mathcal{L}_{w\_denoise}$, and $\mathcal{L}_{con}$ objectives, for enhancing editablity.

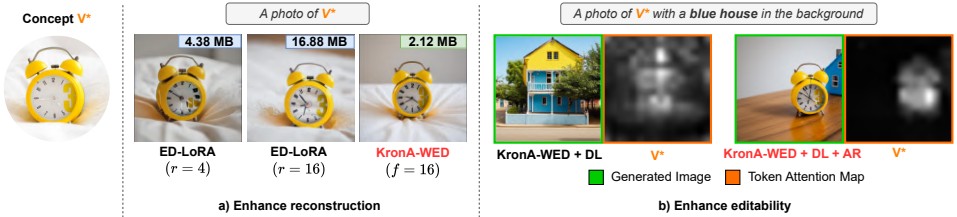

Figure 4: ConceptFlow-S balances the trade-off. (a) KronA-WED adapter can learns complex patterns of concepts with a compact model size. $r$ and $f$ respectively denote LoRA Hu et al. (2024) rank and decomposition factor of KronA (Edalati et al., 2022). (b) Incorporating attention regularization (AR) into disentangled learning (DL) improve the prompt alignment on complex prompts.

where $|| \cdot ||_c$ is the vector-wise norm of a matrix across each column. We use He initialization (He et al., 2015) for $A$ and zero for $B$, while $m$ is initially set to $||W_0||_c$. For the choice of decomposition factors in KronA, we follow the strategy of Yeh et al. (2023) to use a single factor $f$ (see Appendix A.1.3). In Figure 4a, images produced with the KronA-WED module better preserve concept identity than those generated by ED-LoRA (Gu et al., 2024).

**Attention Regularization.** A spread-out attention map with incorrectly activated regions can disrupt the alignment of generated images with intricate prompts (see Figure 4b). Inspired by Avrahami et al. (2023), we propose a novel attention regularization objective to ensure the model reconstructs the learned concept's pixels while focusing the newly-added concept tokens on appropriate image regions. First, following Gu et al. (2024), we present the concept by 2 new tokens $V_{rand}$ (adjective) and $V_{class}$ (noun). For each training input image $\mathbf{x}_i$, we extract the concept foreground mask $M_i$ by using a background removal model BRIA [1]. We then take the average cross-attention map of the concept tokens and penalize their deviation from the extracted masks as follow:

$$\mathcal{L}_{attn} = \lambda_{attn} \sum_{i=1}^{N} \frac{1}{2} \left( ||CA_\theta(V_{rand}, \mathbf{z}_{i,t_i}) \odot (1 - M_i)||_F^2 + ||CA_\theta(V_{class}, \mathbf{z}_{i,t_i}) - M_i||_F^2 \right), \quad (5)$$

where $CA_\theta(V, \mathbf{z}_{i,t_i})$ is the average cross-attention maps between the token $V$ and the noisy latent $\mathbf{z}_{i,t_i}$ of image $\mathbf{x}_i$ at timestep $t_i$ in the U-Net model $\epsilon_\theta$, $\odot$ is the Hadamard product and $|| \cdot ||_F$ is the Frobenius norm. Intuitively, the noun token $V_{class}$ should align with the extracted masks, while the adjective token $V_{rand}$ can activate specific regions within those masks.

It is noteworthy that we only calculate the cross-attention maps in the weak denoising process of disentangled learning (Chen et al., 2024). Figure 4b illustrates the accuracy of cross-attention maps of concept tokens using our attention regularization term, thereby enhancing the interaction between the concept and other subjects in the prompt.

---

[1]https://huggingface.co/briaai/RMBG-1.4

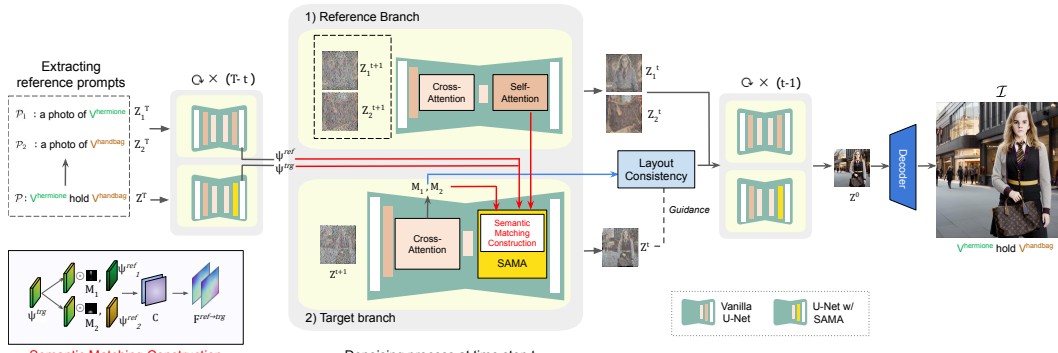

Figure 5: Overview of the composite pipeline in ConceptFlow-M for two concepts. The denoising process has two branches: concepts reference and target. We extract concept tokens in the target prompt $\mathcal{P}$ to form two reference paths. At each timestep, features from the previous timestep and the reference branch are used to enhance the identity details in the target branch through our proposed SAMA module. The latent is updated by layout consistency guidance strategy.

## 4.2 CONCEPTFLOW-M FOR MULTI-CONCEPT COMPOSITION

Given $N$ concept KronA-WEDs $\{\Delta\theta_n\}_{n=1}^N$ and $N$ concept tokens $\{S_n^*\}_{n=1}^N$ ($S_n^* = V_{rand}^n V_{class}^n$), we adopt gradient fusion (Gu et al., 2024) to obtain the fused update weights $\Delta\theta$, thereby forming the fused model $\epsilon_{\theta_0+\Delta\theta}$ with $\theta_0$ is the original weights. Using $\epsilon_{\theta_0+\Delta\theta}$ for generating multi-concept images, we propose **S**ubject-**A**daptive **M**atching **A**ttention (SAMA) module to preserve the identity details of each concept within the composite image, and layout consistency guidance to mitigate the concepts missing issue. The overall pipeline of ConceptFlow-M is illustrated in Figure 5.

### 4.2.1 SUBJECT-ADAPTIVE MATCHING ATTENTION (SAMA)

Although the fused model excels at mimicking the appearance of a single concept in a prompt, there is a degradation in the identity of concepts when multiple concepts are involved (Kong et al., 2024). Therefore, we aim to enhance the identity preservation of multiple concepts within a composite image (*target* branch) by integrating the specific identity details of each concept in images generated from prompts that refer to only a single concept (*reference* branch) throughout the denoising process.

**Extracting reference prompts.** First, a set of newly-added tokens for all concepts in the target prompt $\mathcal{P}$ are extracted, resulting in $K(K \leq N)$ tokens $\{S_k^{\mathcal{P}}\}_{k=1}^K$ corresponding to $K$ reference paths. Each concept $S_k^{\mathcal{P}}$ in the target branch is semantically matched with the reference branch $k$ that generates a single concept image guided by the prompt $\mathcal{P}_k$, defined as "*a photo of $S_k^{\mathcal{P}}$*".

**Semantic matching construction.** Our method builds on the semantic correspondence approach (Truong et al., 2020), tailored with refinements for calculating the matching cost volume and injecting value. Follow Nam et al. (2024), the feature descriptors $\{\psi_k^{ref}\}_{k=1}^K$ in reference branch and $\psi^{trg}$ in target branch from the previous timestep are derived from early decoder layers of the denoising U-Net in Stable Diffusion. To reduce the ambiguity and noise in the matching cost volume resulting from complex spatial features of multiple concepts, we propose a masked matching cost volume $\mathbf{C}_k$ computed for each branch $k$ using the concept foreground mask $\mathbf{M}_k$:

$$\mathbf{C}_k(x,y) = \frac{\psi^{trg} \odot \mathbf{M}_k(x) \cdot \psi_k^{ref}(y)}{\|\psi^{trg} \odot \mathbf{M}_k(x)\| \|\psi_k^{ref}(y)\|}, \tag{6}$$

where $\odot$ denotes the Hadamard product and the foreground mask $\mathbf{M}_k$ is derived from the averaged cross-attention map of the token $S_k^{\mathcal{P}}$, i.e. $V_{rand}^k$ and $V_{class}^k$, at current timestep in the target branch. The estimated semantic correspondence $\mathbf{F}_k^{ref \to trg}$ for the $k$-th concept is then obtained by applying the argmax operation on $\mathbf{C}_k$, as illustrated in Figure 5c. To ensure precise appearance matching for each concept, we calculate the warped reference value $\mathbf{V}_k^{ref \to trg}$ as follow:

$$\mathbf{V}_k^{ref \to trg} = \mathcal{W}(\mathbf{V}_k^{ref}; \mathbf{F}_k^{ref \to trg}) \odot \mathbf{M}_k. \tag{7}$$

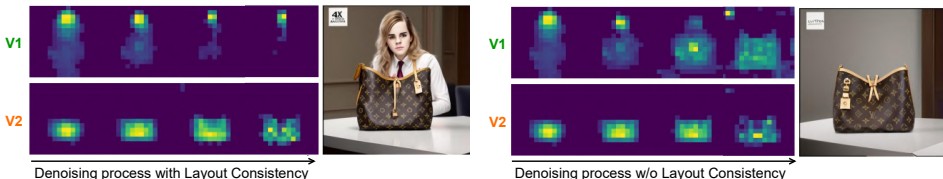

Figure 6: Illustration of layout consistency guidance with the prompt *"$V_1$ advertising $V_2$"*. Ensuring consistency between successive attention maps helps maintain the intended layout of concepts.

These warped reference values are aggregated into a comprehensive value $\mathbf{V}^{\mathcal{W}}$:

$$\mathbf{V}^{\mathcal{W}} = \sum_{k=1}^{K} \mathbf{V}_k^{ref \rightarrow trg} + \mathbf{V}^{trg} \odot \left( 1 - \sum_{k=1}^{K} \mathbf{M}_k \right). \quad (8)$$

Subsequently, SAMA integrates $\mathbf{V}^{\mathcal{W}}$ into the self-attention module in the target branch:

$$\text{SAMA}(\mathbf{Q}^{trg}, \mathbf{K}^{trg}, \mathbf{V}^{\mathcal{W}}) = \text{Softmax}\left( \frac{\mathbf{Q}^{trg}(\mathbf{K}^{trg})^T}{\sqrt{d}} \right) \mathbf{V}^{\mathcal{W}}. \quad (9)$$

In ConceptFlow-M, we apply SAMA to the layers of the middle block and earlier blocks of the decoder in U-Net (see Appendix A.6 for more details).

### 4.2.2 LAYOUT CONSISTENCY GUIDANCE

The concept missing issue in multiple concepts generation is primarily due to the model's difficulty in retaining the layout (Agarwal et al., 2023). As shown in Figure 6, the layout is initially accurate and includes all key concepts at timestep $t = T$, but misaligns by $t = 0$, leading to structural misrepresentation. To mitigate this issue, we propose a test-time layout consistency guidance to maximize the Intersection over Union (IoU) of activated regions of every step consistent with the initial step $T$, thereby strongly encouraging the model to maintain a semantic concept layout.

Specifically, let $\mathbf{A}_k^t(i, j)$ represent the refined activation map at location $(i, j)$ derived from concept token cross-attention map $\mathbf{M}_k^t(i, j)$ at time step $t$ as follow:

$$\mathbf{A}_k^t(i, j) = \begin{cases} \mathbf{M}_k^t(i, j) + \lambda, & \text{if } \mathbf{M}_k^t(i, j) > \tau, \\ \mathbf{M}_k^t(i, j) - \lambda, & \text{if } \mathbf{M}_k^t(i, j) \leq \tau, \end{cases}$$

where $\lambda$ is the adjustment factor and $\tau$ is the threshold that controls whether the value of $\mathbf{M}_k(i, j)$ is enhanced or decreased. The layout consistency loss is formally defined as:

$$\mathcal{L}_{\text{Layout}} = \sum_{k=1}^{K} \left( 1 - \frac{\sum_{i,j} \mathbf{A}_k^t(i, j) \mathbf{A}_k^T(i, j)}{\sum_{i,j} \max(\mathbf{A}_k^t(i, j), \mathbf{A}_k^T(i, j))} \right). \quad (10)$$

At each time step, the latent code $\mathbf{z}_t$ is adjusted based on the direction provided by this layout loss, with a decay factor $\phi_t$ that decreases linearly over time to obtain updated latent code $\mathbf{z}_t'$:

$$\mathbf{z}_t' = \mathbf{z}_t - \phi_t \cdot \nabla_{\mathbf{z}_t} \mathcal{L}_{\text{Layout}}. \quad (11)$$

## 5 EXPERIMENTS

### 5.1 EXPERIMENTAL SETTINGS

**Datasets.** We collect a dataset containing objects, animals and characters, incorporating some sourced from the DreamBench (Ruiz et al., 2023) dataset (see Appendix A.2.1).

**Implementation Details.** For single concept learning, we incorporate KronA-WED to all linear layers in all attention modules of the U-Net with the decomposition factor $f = 16$. We set the weight $\lambda_{attn} = 0.001$ while the other weights are based on disentangled learning settings (Chen et al., 2024). The unified weight $\Delta\theta$ for Concept Flow-M is fused from the two weights learned by ConceptFlow-S with gradient fusion (Gu et al., 2024). More details are provided in Appendix A.2.2.

Table 1: Quantitative comparison between ConceptFlow and other baselines. Bold, underline, and italics indicate the top 1, top 2, and top 3 scores, respectively.

(a) Single concept generation.

| Methods | DINO ↑ | CLIP-T ↑ |
|---|---|---|
| DreamBooth Ruiz et al. (2023) | **0.684** | 0.678 |
| Custom Diffusion Kumari et al. (2023) | 0.503 | **0.784** |
| DisenBooth Chen et al. (2024) | 0.616 | 0.743 |
| ED-LoRA Gu et al. (2024) | 0.667 | 0.703 |
| LoKr Yeh et al. (2023) | *0.679* | 0.688 |
| **ConceptFlow-S** | 0.682 | *0.706* |

(b) Multiple concepts generation.

| Methods | DINO ↑ | CLIP-T ↑ |
|---|---|---|
| Mix-of-Show Gu et al. (2024) | 0.436 | 0.779 |
| Custom Diffusion Kumari et al. (2023) | 0.369 | **0.802** |
| **ConceptFlow-M** | **0.454** | 0.784 |

*A V\* on top of a mirror*

*A V\* sit on the chair*

Figure 7: Qualitative comparison on single concept generation between ConceptFlow-S and other baselines. We focus on the evaluation with complex prompts.

**Evaluation Setting.** In our evaluation process for single concept generation, the prompts are divided into four main types, including Recontextualization, Restylization, Interaction, and Property Modification, and the prompt templates are borrowed from previous works (Ruiz et al., 2023; Gu et al., 2024). For multiple concepts generation, we define a list of prompts for each combination that focuses on the interaction between concepts, especially character-object and character-animal interactions. Detail evaluation setting is provided in Appendix A.2.3.

**Evaluation Metrics.** To evaluate the identity preservation capability, we adopt the DINO score proposed by Ruiz et al. (2023), i.e., the average pairwise cosine similarity between the ViT-S/16 DINO (Caron et al., 2021) embeddings of the generated images and the input real images. The prompt alignment is evaluated by the average cosine similarity between the text prompt and image CLIP (Radford et al., 2021) embeddings. It is noteworthy that we calculate identity preservation separately for each concept in multi-concept images and then average them to get the final score.

**Baselines.** We compare ConceptFlow-S with other joint embedding-weight tuning methods, including DreamBooth (Ruiz et al., 2023), Custom Diffusion (Kumari et al., 2023), DisenBooth (Chen et al., 2024) ED-LoRA (Gu et al., 2024), and LoKr module from LyCORIS (Yeh et al., 2023). For ConceptFlow-M, we compare our method with Mix-of-Show (Gu et al., 2024) and CustomDiffusion (Kumari et al., 2023). Moveover, we do not use regional sampling in Mix-of-Show to ensure fairness in evaluating performance in occlusion scenarios.

## 5.2 COMPARISON BETWEEN CONCEPTFLOW WITH BASELINES

We present the quantitative evaluation results for single concept generation in Table 1a. Our proposed ConceptFlow-S component exhibit comparable results in both metrics, thereby demonstrating its capability in balancing the trade-off of reconstruction and editability. The results of qualitative comparisons are depicted in Figure 7. We provide additional results in Appendix A.3.

For multiple concepts generation, the quantitative evaluation results are depicted in Table 1b. We showcase the qualitative comparisons in Figure 8. ConceptFlow-M demonstrates superior performance in identity preservation compared to other methods. Regarding prompt alignment, Custom

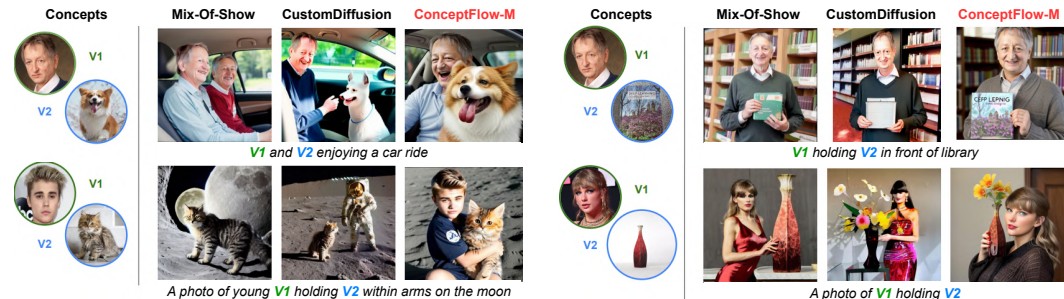

Figure 8: Qualitative comparison between ConceptFlow-M and other baselines.

Table 2: Ablation study for the effectiveness of components in ConceptFlow-S and ConceptFlow-M.

(a) ConceptFlow-S: Kronecker Adapter (KronA), weight decomposition (DORA), and attention regularization (AR).

|  | DINO ↑ | CLIP-T ↑ |
|---|---|---|
| **ConceptFlow-S** | **0.682** | 0.706 |
| W/o KronA | 0.647 (-0.035) | 0.708 (+0.002) |
| W/o DORA | 0.668 (-0.014) | 0.694 (-0.012) |
| W/o AR | 0.660 (-0.022) | **0.710** (+0.004) |

(b) ConceptFlow-M: SAMA, Layout Consistency (LC), and Attention Regularization (AR) in ConceptFlow-S.

|  | DINO ↑ | CLIP-T ↑ |
|---|---|---|
| **ConceptFlow-M** | **0.454** | **0.784** |
| W/o SAMA & LC | 0.435 (-0.019) | 0.778 (-0.006) |
| W/o SAMA | 0.431 (-0.023) | 0.781 (-0.003) |
| W/o LC | 0.442 (-0.012) | 0.775 (-0.009) |
| W/o AR in ConceptFlow-S | 0.440 (-0.014) | 0.767 (-0.017) |

Diffusion (Kumari et al., 2023) scores highest as it generates broad-view images, while our method focus on close-up views, leading to slightly lower scores. Furthermore, we compare ConceptFlow-M with condition-based methods (Gu et al., 2024; Kong et al., 2024) in Appendix A.4.

# 6 ABLATION STUDY

## 6.1 CONCEPTFLOW-S

**Effectiveness of components.** We consistently apply the disentangled learning strategy (Chen et al., 2024) and evaluate the Kronecker Adapter (KronA) (Edalati et al., 2022), weight decomposition (DORA) (Liu et al., 2024), and attention regularization (AR) learning objective. As shown in Table 2a, by substituting the LoRA (Hu et al., 2024) adapter with the KronA (Edalati et al., 2022) adapter, we significantly enhance the reconstruction capability of ConceptFlow-S. Moreover, DORA (Liu et al., 2024) enhances the learning capability of the model, thereby boosting both the DINO and CLIP-T scores. Figure 9a showcases illustrations for these evaluations. We particulary examine the effects of attention regularization (AR) in each prompt category, along with the decomposition factor $f$ in KronA-WED and the number of training images in Appendix A.5.

## 6.2 CONCEPTFLOW-M

**Effectiveness of components.** Table 2b and Figure 9b present the experiments results to demonstrate the effectiveness of SAMA and Layout Consistentcy guidance (LC). Applying SAMA significantly enhances the identity of each concept. However, without LC, one concept is often missed, leading to the issue of semantic matching and a decline in both metrics. Moreover, simply sampling from fused weights can lead to significant identity degradation and missing concepts.

**Utilizing ConceptFlow-S with AR for single concept learning.** In Figure 9b, we display the generated images and the attention map of concept tokens when applying ConceptFlow-M to models learned through ConceptFlow-S, both with and without AR. Withour AR, the maps for both concept tokens are initially unfocused during the early denoising steps, thereby causing our LC to struggle with maintaining the layout in later steps. Consequently, this leads to vague masks for semantic matching, causing a decline in identity preservation and prompt alignment, as indicated in Table 2b.

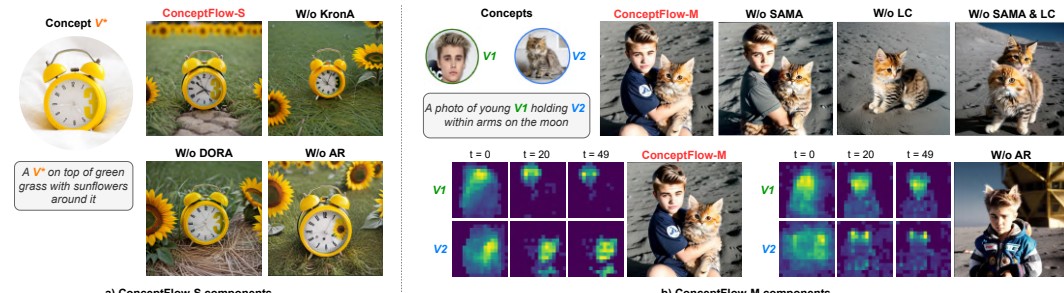

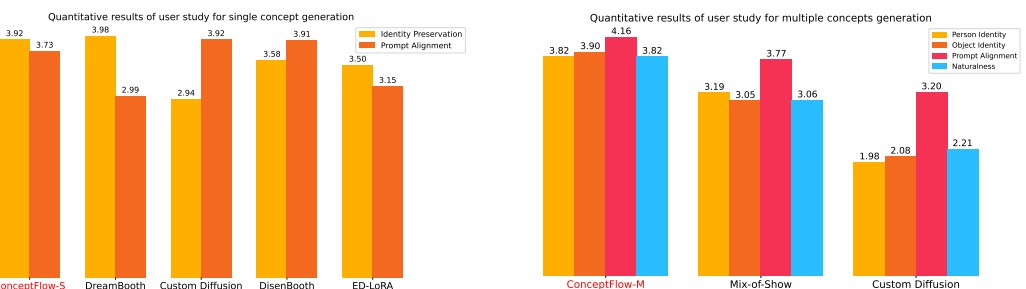

Figure 9: Illustration for the effectiveness of components. (a) ConceptFlow-S: Kronecker Adapter (KronA), weight decomposition (DORA) and attention regularization (AR). (b) ConceptFlow-M: SAMA, Layout Consistency (LC), and utilizing ConceptFlow-S with AR objective.

Figure 10: User study results between ConceptFlow-S and other baselines.

Figure 11: User study results between ConceptFlow-M and other baselines.

## 7 USER STUDY

We conducted a user study to evaluate various methods on a scale from 1 (very bad) to 5 (very good). Follow previous studies (Ruiz et al., 2023; Kumari et al., 2023; Chen et al., 2024), for single concept generation, we considered the metrics of *identity preservation* and *prompt alignment*. Regarding to multiple concepts generation, as the experimental evaluation metrics alone are not sufficiently expressive, we introduced a metric called *naturalness of interaction* to measure how good the naturalness of interaction between the human and object (or animal) in the image is, such as human pose, the size and the position of objects. Detail setup for the user study is provided in Appendix A.7. The results of our study on single concept generation is shown in Figure 10. They indicate that users were satisfied with ConceptFlow-S in terms of both identity preservation (i.e., reconstruction) and prompt alignment (i.e., editability), with average scores of 3.92 and 3.73. For multiple concept generation in Figure 11, ConceptFlow-M outperformed other methods across all metrics by significant margins. Compared to the experiment quantitative results presented in Table 1b, the user study provides deeper insight into the methods' performance in generating multiple concepts.

## 8 CONCLUSION

In this paper, we present ConceptFlow, a robust framework for personalized image generation task. ConceptFlow includes two components: ConceptFlow-S for single concept learning and generation, and ConceptFlow-M for multiple concepts generation. ConceptFlow-S introduces the KronA-WED adapter and a strategy of disentangled learning with attention regularization to balance the trade-off between reconstruction and editability. ConceptFlow-M introduces SAMA and layout consistency guidance to combine concepts from ConceptFlow-S, significantly enhancing the identity of each concept and addressing concept omissions without additional conditions. We demonstrate the effectiveness of ConceptFlow through extensive experiments and user study. ConceptFlow also show its potential in various applications such as advertisement and garment try-on (see Appendix A.8).

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

# A   APPENDIX

This appendix includes our supplementary materials as follow:

- Additional Preliminaries in A.1

- Experiments setting in A.2
- More experiments for ConceptFlow-S in A.3
- More experiments for ConceptFlow-M in A.4
- Additional ablation study for ConceptFlow-S in A.5
- Selective applying SAMA module for CoceptFlow-M in A.6
- Detail user study setup and results in A.7
- Potential applications of ConceptFlow in A.8
- Limitation and future work in A.9

## A.1 ADDITIONAL PRELIMINARIES

### A.1.1 FINE-TUNING STABLE DIFFUSION FOR SINGLE CONCEPT GENERATION

Denote the small set of images of the specific concept $s$ as $\mathbb{C}_s = \{\mathbf{x}_i\}_{i=1}^N$, where $\mathbf{x}_i$ is the $i^{th}$ image and $N$ is the number of images (usually from 3 to 5). Single concept generation methods (Gal et al., 2023; Ruiz et al., 2023; Kumari et al., 2023; Gu et al., 2024) focus on binding a newly-added text token $V_s$ to the concept $s$ with the following fine-tuning objective:

$$\min \ \mathbb{E}_{\mathbf{z}=\mathcal{E}(\mathbf{x}),\mathbf{x}\sim\mathbb{C}_s,\boldsymbol{\epsilon},t}\left[||\boldsymbol{\epsilon} - \boldsymbol{\epsilon}_\theta(\mathbf{z}_t, t, E_T(V_s))||_2^2\right], \tag{12}$$

where $\boldsymbol{\epsilon}_\theta$ is the U-Net based conditional diffusion model, $E_T$ is the CLIP (Radford et al., 2021) text encoder and $\mathcal{E}$ is the VAE encoder. Different methods use the objective in Eq. 12 to fine-tune different parameters. For example, Textual Inversion (Gal et al., 2023) fine-tunes the embedding of $V_s$ in the CLIP text encoder, DreamBooth (Ruiz et al., 2023) jointly fine-tunes the entire U-Net model and the embedding of $V_s$.

### A.1.2 LOW RANK ADAPTATION (LoRA)

Low-rank adaptation (LoRA) (Hu et al., 2024) was originally introduced to adapt large language models for downstream tasks. Based on the assumption that updates made during the fine-tuning exhibit a low intrinsic rank, LoRA proposes using the product of two low-rank matrices to gradually update the pre-trained weights, thereby significantly reduces the number of trainable parameters.

For a pre-trained weight matrix $W_0 \in \mathbb{R}^{d \times k}$, LoRA models the weight update $\Delta W \in \mathbb{R}^{d \times k}$ utilizing a low-rank decomposition, expressed as $BA$, where $B \in \mathbb{R}^{d \times r}$ and $A \in \mathbb{R}^{r \times k}$, with $r \ll \min(d, k)$. Then, the fine-tuned weight $W'$ can be formulated as:

$$W' = W_0 + \Delta W = W_0 + \underline{BA}, \tag{13}$$

where the underlined parameters are being trained and $W_0$ is frozen during the fine-tuning process. At the start of training, $A$ is initialized with a uniform Kaiming distribution and $B$ is initially set to zero, which leads to $\Delta W = BA$ bezing zero. As we can merge the updated weights $\Delta W$ with $W_0$ to obtain $W'$ before performing inference, LoRA does not introduce any extra latency compared to the original model.

### A.1.3 KRONECKER ADAPTER (KRONA)

Despite being efficient, LoRA (Hu et al., 2024) can suffers from a performance drop compared to the full fine-tuning because of the strong assumption imposed by its low-rank structure for task-specific updates. This behavior can be explained based on the number of singular vector of the updated matrices, where matrices with higher number of singular vectors might have better capability in capturing the expresitivy (Hu et al., 2024). Kronecker product decomposition is an alternative factorization method that does not depend on the low-rank assumption. By adopting this decomposition to the fine-tuning process, Kronecker Adapter (KronA) (Edalati et al., 2022) can improve the performance of large language models in specific tasks without increasing the inference latency compared to LoRA.

A core component of Kronecker decomposition is Kronecker product ($\otimes$), which is a matrix multiplication method that allows multiplication between matrices of different shapes. Given two input

Table 3: Number of parameters of LoRA (Hu et al., 2024) and KronA (Edalati et al., 2022) adapters.

| Adapter | Decomposed Matrices | Number of Params |
|---------|---------------------|------------------|
| LoRA | $A \in \mathbb{R}^{r \times k}, B \in \mathbb{R}^{d \times r}$ $r \ll \min(d, k)$ | $r(d + k)$ |
| KronA | $A \in \mathbb{R}^{a_1 \times a_2}, B \in \mathbb{R}^{b_1 \times b_2}$ $a_1 \times b_1 = d, a_2 \times b_2 = k$ | $a_1 a_2 + b_1 b_2$ |

matrices $A \in \mathbb{R}^{a_1 \times a_2}$ and $B \in \mathbb{R}^{b_1 \times b_2}$, Kronecker product results a matrix $W \in \mathbb{R}^{a_1 b_1 \times a_2 b_2}$. We can view the matrix $W$ as $a_1 \times a_2$ blocks, where the block $(i, j)$ is equal to the multiplication of the element $A_{i,j}$ and the matrix $B$. Hence, $W$ can be formulated as:

$$W = A \otimes B = \begin{bmatrix} A_{1,1}B & \cdots & A_{1,a_2}B \\ \vdots & \ddots & \vdots \\ A_{a_1,1}B & \cdots & A_{a_1,a_2}B \end{bmatrix}. \tag{14}$$

Utilizing Kronecker product, Edalati *et al.* (Edalati et al., 2022) replaces low-rank decomposition in LoRA with Kronecker decomposition to develop the Kronecker Adapter (KronA). The difference in structure between LoRA (Hu et al., 2024) and KronA (Edalati et al., 2022) is shown in Figure **??**. Specifically, with a pre-trained weight $W_0 \in \mathbb{R}^{d \times k}$, the updated weight $\Delta W$ is expressed as $A \otimes B$, where $A \in \mathbb{R}^{a_1 \times a_2}$ and $B \in \mathbb{R}^{b_1 \times b_2}$, with $a_1 \times b_1 = d$ and $a_2 \times b_2 = k$. Consequently, we can obtain the fine-tuned weight $W'$ as:

$$W' = W_0 + \Delta W = W_0 + A \otimes B. \tag{15}$$

An important features of Kronecker product is that the rank of result matrix is not depend on the two input matrics, thereby making KronA (Edalati et al., 2022) suitable for parameter-efficient fine-tuning (PEFT). Denoting $\Delta W_{\text{LoRA}}$ and $\Delta W_{\text{KronA}}$ as the obtain updated weights from LoRA (Hu et al., 2024) and KronA (Edalati et al., 2022) fine-tuning, and $rank(\cdot)$ is the rank of a matrix. We can bound the rank of $\Delta W_{\text{LoRA}}$ and $\Delta W_{\text{KronA}}$ as follow:

$$rank(\Delta W_{\text{LoRA}}) \leq \min(r(A), r(B)) \leq r, \tag{16}$$

$$rank(\Delta W_{\text{KronA}}) = r(A) \cdot r(B) \leq \min(a_1, a_2) \cdot \min\left(\frac{d}{a_1}, \frac{k}{a_2}\right). \tag{17}$$

Therefore, the values of the decomposition factors $a_1$ and $a_2$ sorely influence the number of parameters in the adapter. KronA (Edalati et al., 2022) has the potential to achieve an updated matrix rank comparable to that of full fine-tuning methods.

For the choice of decomposition factors (*i.e.* hyperparameters) $a_1$ and $a_2$ in KronA adapters in our thesis, we follow the strategy of Yeh *et al.* (Yeh et al., 2023). Specifically, we reduce them to a single decomposition factor $f$, and the value of $a_1, b_1$ will be specified based on $d$ as follow (similar to $a_2$, $b_2$ and $k$) :

$$a_1 = \max(u \leq \min(f, \sqrt{d}) \mid d \bmod u = 0), \quad b_1 = \frac{d}{a_1} \tag{18}$$

The number of parameters of LoRA (Hu et al., 2024) and KronA (Edalati et al., 2022) are depicted in Table 3.

### A.1.4 WEIGHT-DECOMPOSED LOW-RANK ADAPTATION (DORA)

Through the weight decomposition analysis to investigate the inherent differences between fine-tuning (FT) and LoRA (Hu et al., 2024), DORA (Liu et al., 2024) decomposes the pre-trained weight into two components, magnitude and direction, for fine-tuning, and specifically employing LoRA for directional updates to efficiently minimize the number of trainable parameters. DORA enhances the learning capacity and training stability of LoRA without introducing any additional inference overhead.

Specifically, denote the original weight matrix as $W_0 \in \mathbb{R}^{d \times k}$, the weight decomposition of $W_0$ is formulated as:

$$W_0 = m \frac{V}{||V||_c} = ||W_0||_c \frac{W_0}{||W_0||_c}, \tag{19}$$

where $m \in \mathbb{R}^{1 \times k}$ is the magnitude vector, $V \in \mathbb{R}^{d \times k}$ and $|| \cdot ||_c$ is the vector-wise norm of a matrix across each column. Note that each column of $V/||V||_c$ is a unit vector, and this term is called direction component. Consequently, DORA formulated the fine-tuned weight $W'$ as:

$$W' = \underline{m} \frac{V + \underline{\Delta V}}{||V + \underline{\Delta V}||_c} = \underline{m} \frac{W_0 + \underline{BA}}{||W_0 + \underline{BA}||_c}, \tag{20}$$

where $\Delta V$ is the incremental directional update learned by multiplying two low-rank matrices $B \in \mathbb{R}^{d \times r}$ and $A \in \mathbb{R}^{r \times k}$, and the underlined parameters denote the trainable parameters. The initialization of $m$, $B$ and $A$ ensures that $W'$ equal to $W_0$ before the fine-tuning. Specifically, we assign $m = ||W_0||_c$, $B$ and $A$ are initialized in the same way with LoRA (Hu et al., 2024).

### A.1.5 DISENTANGLED LEARNING.

Chen et al. (2024) proposed learning separate textual and visual embeddings to prevent the mixing of a concept's identity with irrelevant details such as background or the pose during fine-tuning. Specifically, to extract the identity-irrelevant embedding of image $x_i$, they adopt the pretrained CLIP (Radford et al., 2021) image encoder $E_I$ to obtain image features $f_i^{(p)} = E_I(x_i)$, then filtering out the identity information from $f_i^{(p)}$ by a learnable mask $M$. These features are consequently fed into an adapter $MLP$ with skip connection to be tranformed into the same space as text feature $f_s$ as follow:

$$f_i = M * f_i^{(p)} + MLP(M * f_i^{(p)}), \; i = 1, 2, \cdots N, \tag{21}$$

where $N$ is the number of training images.

The learning objectives of DisenBooth (Chen et al., 2024) includes a denoising objective $\mathcal{L}_{denoise}$ (similar to Eq. 12) and two disentangled objectives, which are weak denoising objective $\mathcal{L}_{w-denoise}$ and contrastive embedding objective $\mathcal{L}_{con}$. Firstly, the main objective of fine-tuning process $L_{denoise}$ is defined as:

$$\mathcal{L}_{denoise} = \sum_{i=1}^{N} ||\boldsymbol{\epsilon}_i - \boldsymbol{\epsilon}_\theta(\mathbf{z}_{i,t_i}, t_i, f_i + f_s)||_2^2. \tag{22}$$

The weak denoising objective $L_{w-denoise}$ helps to learn meaningful text features $f_s$ that capable of capturing the identity-relevant information, *i.e.* the common part of the training images:

$$\mathcal{L}_{w-denoise} = \lambda_{w-denoise} \sum_{i=1}^{N} ||\boldsymbol{\epsilon}_i - \boldsymbol{\epsilon}_\theta(\mathbf{z}_{i,t_i}, t_i, f_s)||_2^2. \tag{23}$$

Moreover, since we expect $f_s$ and $f_i$ to capture disentangled information of the image $x_i$, the embeddings $f_s$ and $f_i$ should be contrastive and their similarities are expected to be low. Therefore, the contrastive embedding objective is added as follow:

$$\mathcal{L}_{con} = \lambda_{con} \sum_{i=1}^{N} cos(f_s, f_i), \tag{24}$$

where $cos(\cdot)$ is the cosine similarity between two vectors.

### A.2 EXPERIMENTS SETTING

### A.2.1 DATASET

To evaluate our proposed method, we collect a dataset containing objects, animals and characters, incorporating some sourced from the DreamBench (Ruiz et al., 2023) dataset. Our dataset including 12 objects, 5 animals and 7 characters (5 *regular humans* and 2 *special humans*), which are shown in Figure 12. The key distinction between regular humans and special humans is that for the latter, we aim to also preserve their outfits, while for regular human characters, our primary focus is only on their faces. For multiple concepts generation, we focus on the interaction between character-object and character-animal, results in 60 combinations.

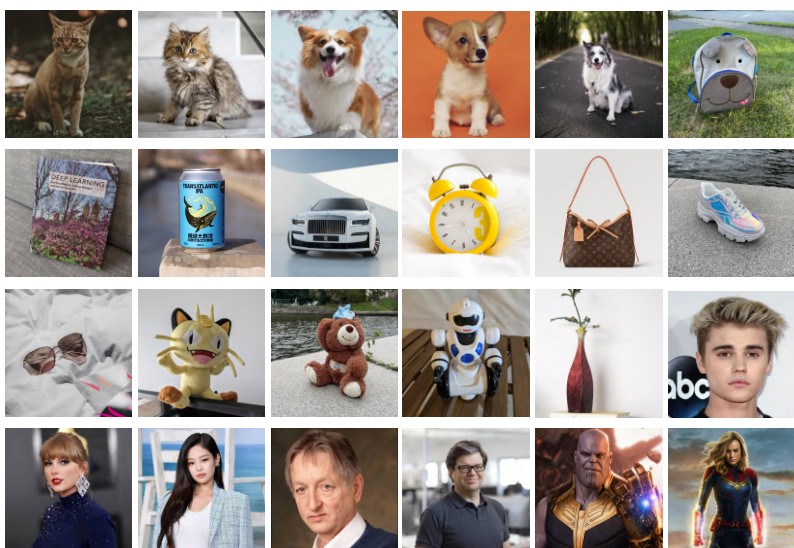

Figure 12: All concepts in our dataset for experiments.

### A.2.2 IMPLEMENTATION DETAILS

**Single Concept Generation.** In our experiments, we incorporate KronA-WED to all linear layers in all attention modules of the U-Net with the decomposition factor $f = 16$. We use the AdamW (Loshchilov & Hutter, 2019) optimizer with a learning rate 1e-3, 5e-4 and 1e-3 for tuning text embedding, U-Net and image feature adapter, respectively. The fine-tuning iterations are 2,000 for all concepts. We set the weight of each fine-tuning objective as follow: $\lambda_{w-denoise} = \lambda_{con} = 0.001$ (similar to DisenBooth (Chen et al., 2024)), and $\lambda_{attn} = 0.001$. The pretrained checkpoints for Stable Diffusion we adopt for all methods is Realistic Vision V5.1 [2] as it well-known high quality outputs for realistic images. The sampling processes are performed with DPM-Solver (Lu et al., 2022) scheduler with 50 sampling steps.

**Multiple Concepts Generation.** For gradient fusion (Gu et al., 2024), we use the LBFGS optimizer (Liu & Nocedal, 1989) with 500 and 50 steps to optimize the text encoder and Unet, respectively. In our experiment, the unified weight $\Delta\theta$ for Concept Flow-M is fused from the two weights learned by ConceptFlow-S. The sampling process is performed using the DPM-Solver scheduler (Lu et al., 2022) with 50 sampling steps. The matching process with SAMA module starts from step 4 of denoising. For layout consistency guidance, we set refined threshold $\tau$ to 0.55, adjustment factor $\lambda$ to 0.5, and the decay factor $\phi$ is set to 10.0

### A.2.3 EVALUATION SETTING

**ConceptFlow-S.** In our evaluation process for each concept within the categories of objects, characters, and animals, we employ 25, 20, and 20 prompts respectively. These prompts are divided into four main types, including Recontextualization, Restylization, Interaction, and Property Modification, and the prompt templates are borrowed from previous work (Ruiz et al., 2023; Gu et al., 2024) and they are depicted in Figure 13. Moreover, for the object category, we skip the Restylization and place a greater emphasis on assessing the reconstruction and editability of various methods in intricate scenarios. Consequently, we utilize a higher number of prompts related to recontextualization and interaction. We sample 25 images for each prompt with fixed random seeds for the reproducibility. The sampling processes are performed with DPM-Solver (Lu et al., 2022) scheduler with 50 sampling steps.

**ConceptFlow-M.** We define a list of prompts for each combination that focuses on the interaction between subjects. For example, in the character-object type, our prompts center around verb actions (*e.g.*, hold, wear), while in the character-animal type, verbs are centered around scenarios where

---

[2]https://civitai.com/models/4201?modelVersionId=130072

| | Prompt for objects | Prompt for characters | Prompt for animals |
|---|---|---|---|
| **Recontextualization** | a <TOK> in the jungle
a <TOK> in the snow
a <TOK> on the beach
a <TOK> on a cobblestone street
a <TOK> with a city in the background
a <TOK> with a mountain in the background
a <TOK> with a blue house in the background
a <TOK> with a wheat field in the background
a <TOK> with a tree and autumn leaves in the background
a <TOK> with the Eiffel Tower in the background | A photo of <TOK> on the beach, small waves, detailed symmetric face, beautiful composition
A <TOK>, in front of Eiffel tower
A <TOK>, near the mount fuji
A <TOK>, in the forest
A <TOK>, walking on the street | A <TOK> in the swimming pool
A <TOK> in front of Eiffel tower
A <TOK> near the mount fuji
A <TOK> in the forest
A <TOK> walking on the street |
| **Restylization** | | A <TOK>, cyberpunk 2077, 4K, 3d render in unreal engine
A watercolor painting of a <TOK>
A painting of a <TOK> in the style of Vincent Van Gogh
A painting of a <TOK> in the style of Claude Monet
A <TOK> in the style of Pixel Art | A <TOK> cyberpunk 2077, 4K, 3d render in unreal engine
A watercolor painting of a <TOK>
A painting of a <TOK> in the style of Vincent Van Gogh
A painting of a <TOK> in the style of Claude Monet
A <TOK> in the style of Pixel Art |
| **Interaction** | a <TOK> floating on top of water
a <TOK> floating in an ocean of milk
a <TOK> on top of pink fabric
a <TOK> on top of a wooden floor
a <TOK> on top of green grass with sunflowers around it
a <TOK> on top of a mirror
a <TOK> on top of the sidewalk in a crowded street
a <TOK> on top of a dirt road
a <TOK> on top of a white rug
a <TOK> on top of a purple rug in a forest | A <TOK> sit on the chair
A <TOK> ride a horse
A <TOK>, wearing a headphone
A <TOK>, wearing a sunglass
A <TOK>, wearing a Santa hat | A <TOK> sit on the chair
A <TOK> on the boat
A <TOK> wearing a headphone
A <TOK> wearing a sunglass
A <TOK> playing with a ball |
| **Property change** | a red <TOK>
a purple <TOK>
a shiny <TOK>
a wet <TOK>
a cube shaped <TOK> | A smiling <TOK>
An angry <TOK>
A running <TOK>
A jumping <TOK>
A <TOK> is lying down | A sad <TOK>
An angry <TOK>
A running <TOK>
A jumping <TOK>
A <TOK> is lying down |

Figure 13: Our evaluation prompt for single concept generation.

humans interact with animals (*e.g.*, play, hold within arms). We sample 10 images for each prompt, resulting in an average of 60 images for each combination.

## A.3 MORE QUALITATIVE COMPARISONS

We provide additional qualitative comparisons of ConceptFlow-S and ConceptFlow-M with other baselines in Figure 14 and Figure 15, respectively.

## A.4 COMPARISONS BETWEEN CONCEPTFLOW-M WITH CONDITION-BASED METHODS

Multi-concept condition-based methods (Kong et al., 2024; Gu et al., 2024), which rely on extra inputs like masks, bounding boxes, and pose conditions, often struggle in interaction scenarios where subjects occlude each other. In such cases, these approaches may not properly handle occlusion. We present additional qualitative results in Figure 16 where we compare our ConceptFlow-M with OMG (Kong et al., 2024) and Mix-of-Show (Gu et al., 2024) with regional sampling. OMG attempts to address occlusion by first generating a layout image, then blending noise into specific regions using predicted masks, while Mix-of-Show manipulates cross-attention layers guided by bounding boxes. Although Mix-of-Show generally ensures that all concepts are present, it fails to maintain natural interactions between subjects. In scenarios where subjects significantly occlude each other (e.g., a person wearing glasses), it cannot preserve all concepts effectively. A major limitation in OMG is the mismatch between predicted masks in the layout generation stage and the actual shape of the subjects, leading to significant identity loss and inaccurate region blending.

## A.5 ADDITIONAL ABLATION STUDY FOR CONCEPTFLOW-S

### A.5.1 ATTENTION REGULARIZATION (AR) OBJECTIVE

Table 4 indicates that AR improves prompt alignment for recontextualization and interaction but is not well-suited for prompts involving interaction and property changes. Figure 17 showcases the illustrations for these evaluations.

The effect of attention regularization (AR) on restylization and property change prompts is due to the attention map of the style (or property) token being overshadowed by the attention of the concept token in regions containing the concept within the image. As a result, the expected style (or property) is not visible in the concept region. Despite this limitation, we use AR in ConceptFlow-S

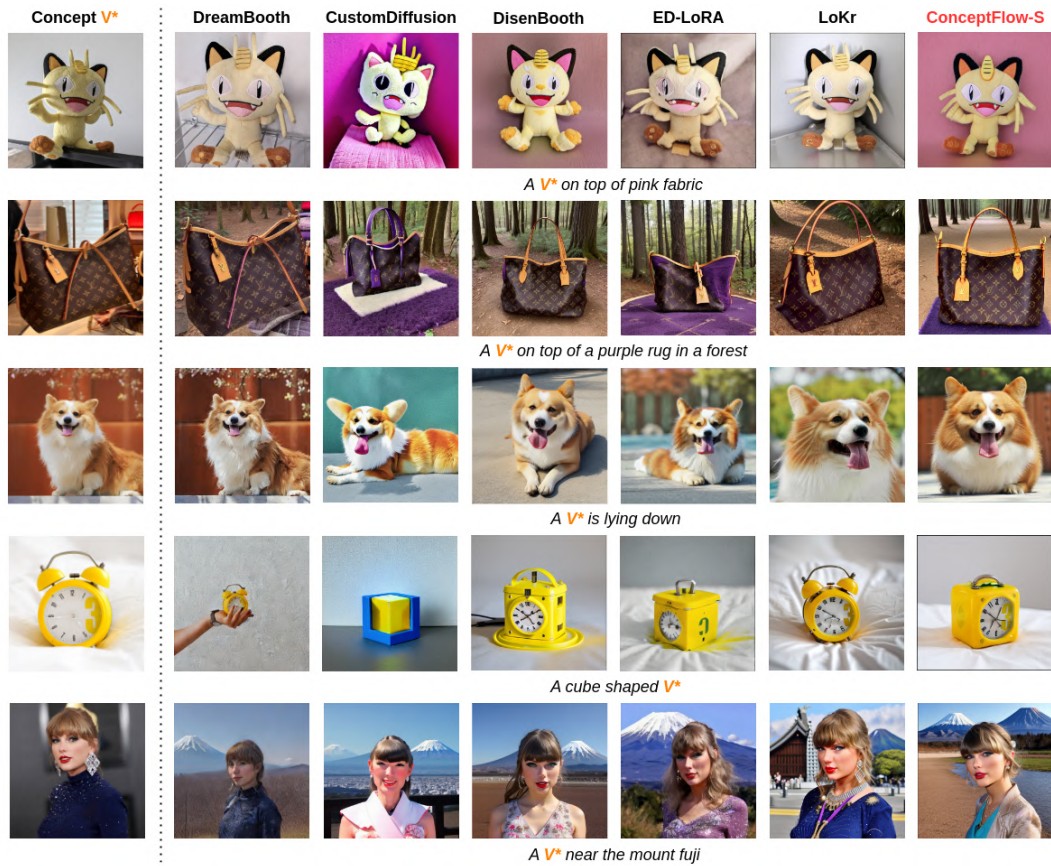

Figure 14: Qualitative comparison on single concept generation between ConceptFlow-S and other baselines. We focus on the evaluation with complex prompts.

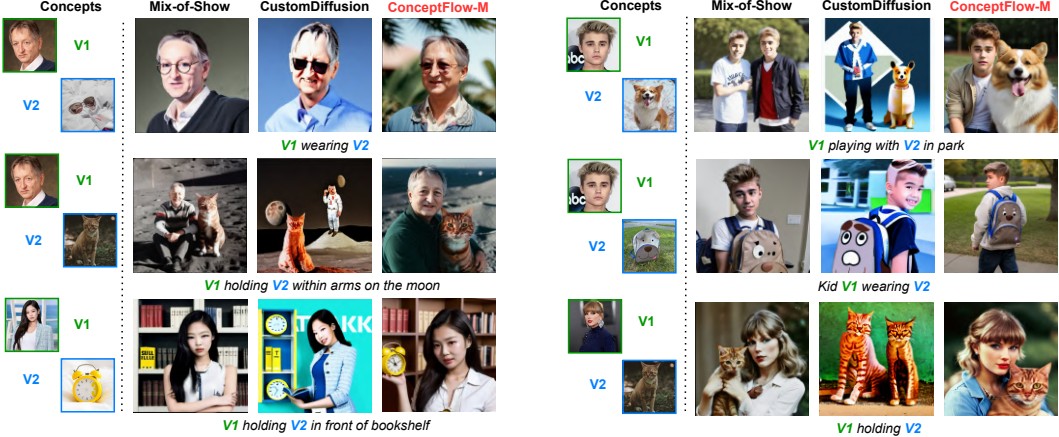

Figure 15: Qualitative comparison on multiple concepts generation between ConceptFlow-M and other baselines.

because it supports the generation of multiple concepts in ConceptFlow-M, where effective handling of interaction and recontextualization prompts is essential.

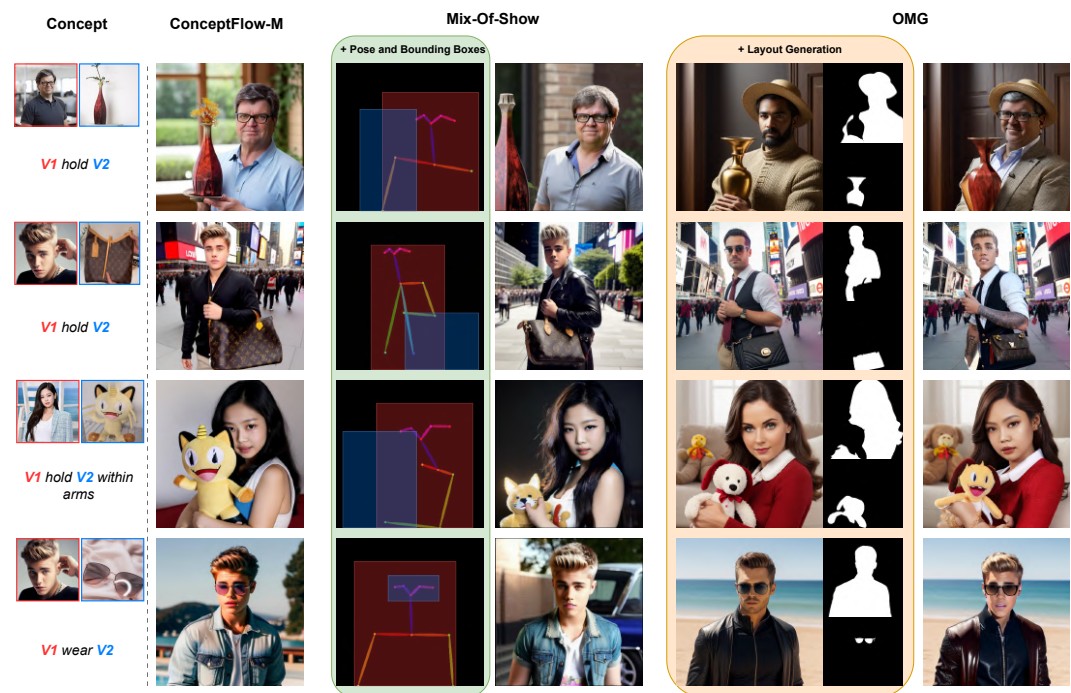

Figure 16: Qualitative comparison between ConceptFlow-M, which operates solely from text prompts, and other methods that use additional conditions. Mix-of-Show (Gu et al., 2024) employs regional sampling with pose and bounding box inputs, while OMG (Kong et al., 2024) is a blend-based approach that requires a generated layout for precise blending of concept noise.

Table 4: CLIP-T score of CopceptFlow-S with and without AR in each prompt category. In each row, the value in bold is the highest score.

|  | Our | W/o AR |
|---|---|---|
| Recontextualization | **0.756** | 0.752 (-0.004) |
| Restylization | 0.678 | **0.708** (+0.03) |
| Interaction | **0.711** | 0.701 (-0.01) |
| Property change | 0.642 | **0.658** (+0.016) |

### A.5.2 KRONECKER DECOMPOSITION FACTOR

Despite the fact that the rank of updated matrices obtained from KronA (Edalati et al., 2022) adapter is not depend on the factorization matrices, choosing the right value for decomposition factor $f$ (Yeh et al., 2023) is an important detail. According to Table 3, the minimum number of paramerers in a KronA adapter is as follow:

$$a_1 a_2 + b_1 b_2 = a_1 a_2 + \frac{d}{a_1} \frac{k}{a_2} \geq 2\sqrt{dk}, \tag{25}$$

and the equation holds when $a_1 = \sqrt{d}$ and $a_2 = \sqrt{k}$ and $f \geq \max(\sqrt{d}, \sqrt{k})$ (following Equation 18).

Increasing the value of $f$ results in fewer parameters and vice versa. This variation may cause issues of underfitting (*i.e.* insufficient to capture the fine-grained details) or overfitting to the training images, which are illustrated in Figure 18. In all of our experiments, we set the value $f$ to 16 to balance the efficiency and the effectiveness of ConceptFlow-S.

### A.5.3 NUMBER OF TRAINING IMAGES.

We present the generation results of ConceptFlow-S in scenarios involving both a single training image and multiple training images (ranging from 3 to 5 images used in our experiments) in Figure 19.

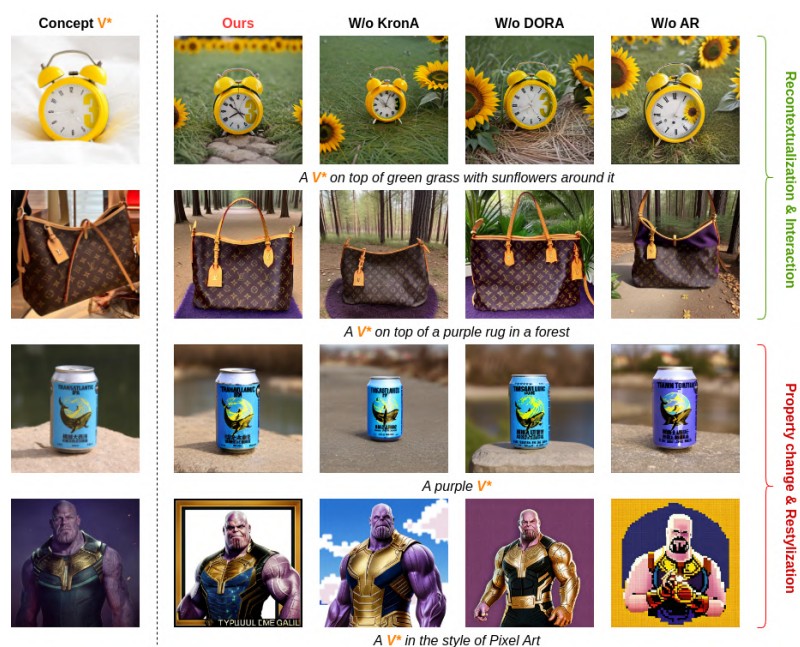

Figure 17: Illustration for the effectiveness of each component in ConceptFlow-S, including Kronecker Adapter (KronA), weight decomposition (DORA) and attention regularization (AR). In this figure, "W/o" stands for "Without".

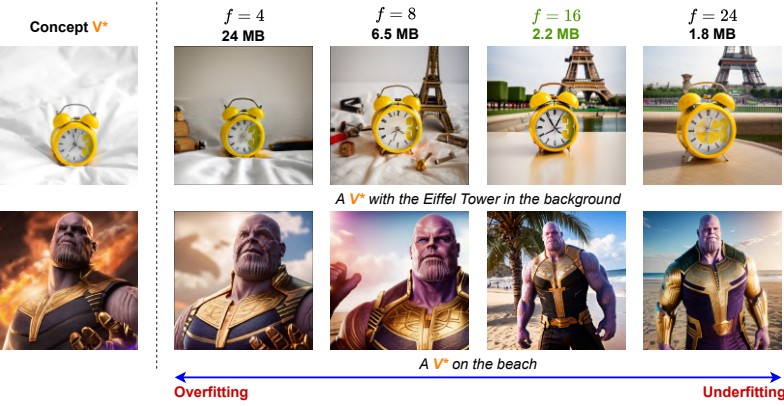

Figure 18: The effect of decomposition factor $f$ in KronA (Edalati et al., 2022) adapter for the concept learning and generation outputs. We provide the model size below each factor value.

For concepts that do not contain many complex details such as regular humans and animals, the results indicate that ConceptFlow-S can generate desirable images with just one training image. However, in this setting, the model tends to overfit the training image to preserve the concept's identity. Moreover, the diversity of the outcomes may decrease due to overfitting to the concept layout in one training image setting.

## A.6 SELECTIVE APPLYING SAMA MODULE IN CONCEPTFLOW-M

Experimental results demonstrate that substituting all self-attention modules with the SAMA module does not yield satisfactory results due to the layer-specific sensitivity of features. Since the estimated correspondence $\mathbf{F}_k^{ref \to trg}$ is calculated from the cost volume $\mathbf{C}_k$ derived from early decoder layers, which focus on semantics and structures (Tumanyan et al., 2023; Zhang et al., 2024; Mou et al., 2024), SAMA should be applied to layers where spatial features have significant semantic

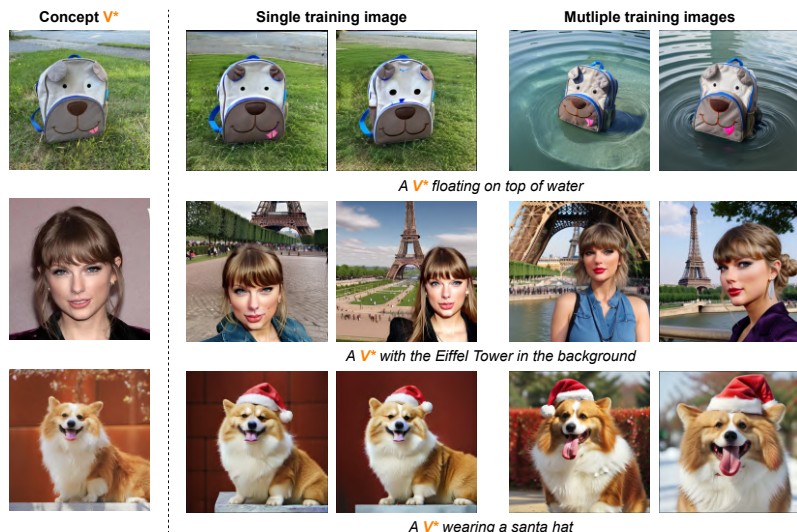

Figure 19: Generation of single concept using ConceptFlow-S in different number of training images. For multiple images, we use from 3-5 images in our experiments.

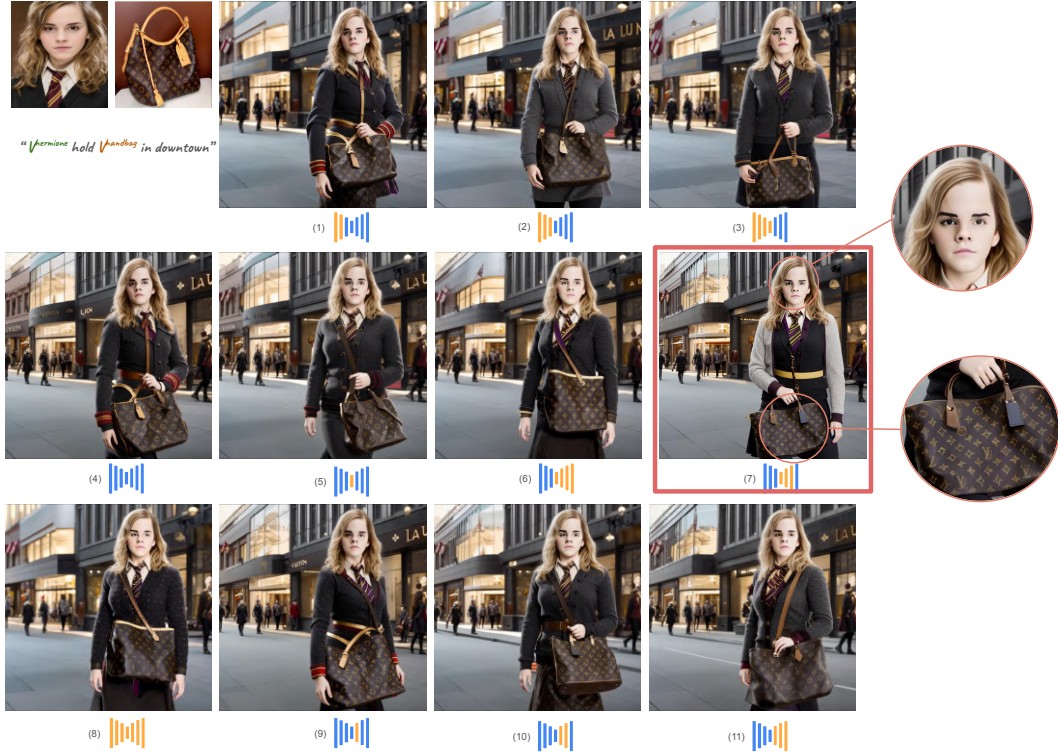

Figure 20: The impact of the SAMA module when implemented across various blocks in a U-Net architecture. The blue color signifies the original basic block, while the yellow color denotes the basic block in which self-attention is substituted by SAMA.

appearance information. Through empirical observation, we achieve superior results when applying SAMA to the layers of the middle block and earlier blocks of the decoder in U-Net (Ronneberger et al., 2015) instead of replacing all self-attention modules in the 7 basic blocks with SAMA., as shown in Figure 20.

### A.7 USER STUDY

#### A.7.1 PARTICIPANTS

We invited 20 participants (16 males and 4 females) with age range from 18 to 25 from our research community and university to participate in the study. Half of them have a background in AI, and some are acquainted with our evaluation metrics. They brought diverse perspectives to the evaluation process, ensuring an objective assessment.

#### A.7.2 STUDY SETUP

We utilized Stable Diffusion 1.5 with Realistic Vision checkpoint for all methods. To ensure objectivity, we blinded the method so that participants did not know which method the image belonged to. The reference image of concepts and the images generated by methods were presented side-by-side for evaluation.

For single concept generation, we created 4 batches corresponding to 4 prompt categories: recontextualization, interaction, restylization, and property change. Each batch contains 24 prompts with a total of 120 images. The participants engaged randomly in one of these batches with the shuffling questions. We evenly distributed the assignments across 4 batches to achieve a comprehensive evaluation. Furthermore, we excluded LoKr (Yeh et al., 2023) from our user study comparisons because its trade-off between reconstruction and editability is similar to that of DreamBooth (Ruiz et al., 2023) (refer to Table 1a), both of which maintain high identity preservation but substantially reduce prompt alignment.

Meanwhile, in multiple concept generation, we created 2 batches corresponding to two categories of combination: human-object and human-animal. Each batch contains 30 prompts with a total of 90 images. Similar to the single concept study, one of these combination types will be randomly assigned to each participant.

#### A.7.3 EVALUATION METRICS

The participants were instructed to rate the performance of various methods on a scale from 1 (very bad) to 5 (very good). To ensure fair and consistent evaluations, we provided a reference document outlining the criteria for each score. Follow previous studies (Ruiz et al., 2023; Kumari et al., 2023; Gu et al., 2024), for single concept generation, we considered the metrics of identity preservation and prompt alignment. Regarding to multiple concepts generation, we add a metric called *naturalness of interaction* to measure how good the naturalness of interaction between the human and object (or animal) in the output image is, such as human pose, the size and the position of objects. Detail setup for the user study is provided in Appendix A.7.

#### A.7.4 QUANTITATIVE RESULTS

The results of our study on single concept generation are displayed in Figure 21. They indicate that users were satisfied with ConceptFlow-S in terms of identity preservation (i.e., reconstruction) and prompt alignment (i.e., editability) with the average scores of 3.92 and 3.73.

For multiple concept generation, the results are depicted in Figure 22 which showed that ConceptFlow-M outperformed other methods in all metrics with large margins, thereby demonstrating the effectiveness this component. Compared to the experiment quantitative results presented in Table 1b, the user study provides deeper insight into the methods' performance in generating multiple concepts, as the experimental evaluation metrics alone are not sufficiently expressive.

### A.8 POTENTIAL APPLICATIONS

By generating realistic images of clothing items on various body types and poses, it becomes possible to create virtual fitting rooms where customers can visualize how different garments will look on them without physically trying them on. This application can significantly enhance the online shopping experience, reducing return rates and increasing customer satisfaction. Moreover, the personalized image generation capabilities of our framework ConceptFlow offer significant potential in the field of advertisement. By enabling the creation of high-quality customized concept models,

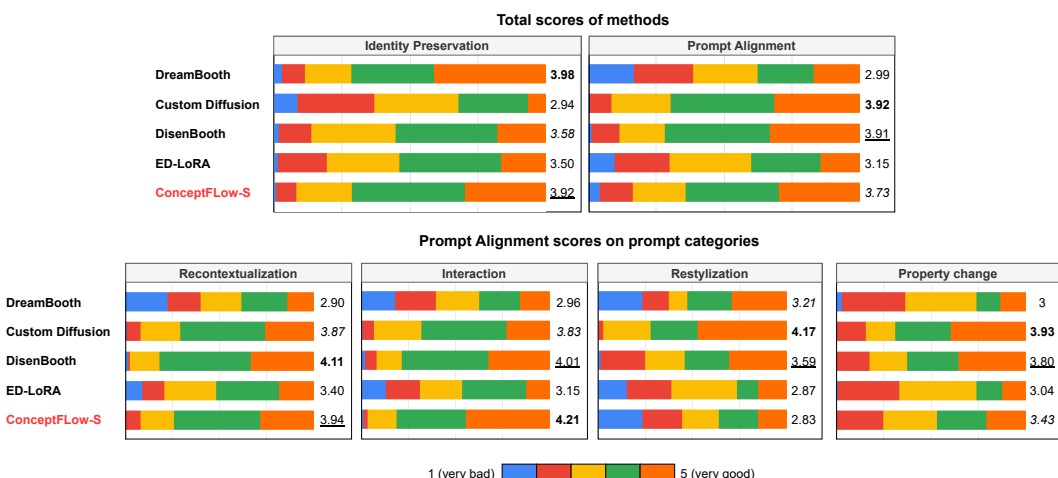

Figure 21: Quantitative results of our user study for single concept generation. The values in bold, underline, and italic indicate the top 1, top 2, and top 3 scores, respectively.

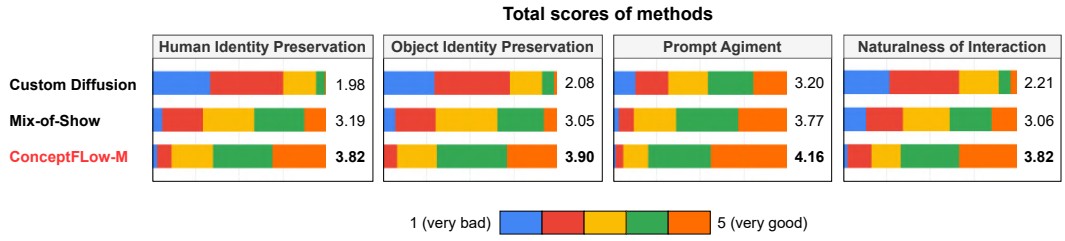

Figure 22: Quantitative results of our user study for multiple concepts generation. The values in bold indicate the highest scores.

businesses can generate compelling visual content tailored to specific products. We showcases these applications of ConceptFlow in Figure 23.

## A.9 LIMITATIONS AND DISCUSSION

Firstly, since ConceptFlow framework is built on top of Stable Diffusion 1.5 model, it inherits the limitations of the pre-trained Stable Diffusion such as bad hand drawings for human concepts.

In the realm of single concept generation, our ConceptFlow-S component successfully balances reconstruction and editability, but enhancing both capabilities simultaneously remains a challenge, particularly in terms of prompt restylization and property changes (as discussed in Appendix A.5.1). Furthermore, capturing sophisticated details of concepts like text, logos, and other elements remains demanding. As illustrated in Figure 24, ConceptFlow-S struggles to accurately retain these intricate details. The issue may primarily arise from the resolution set to 512 for training and generation in our experiments. When we increased this resolution to 768, there was a slight improvement in the quality of the text region, but it still falls short compared to the reference images. We also investigated the output of DreamBooth (Ruiz et al., 2023) and observed the similar behaviors, except on the generated images that are almost identical to the training images (*i.e.* overfitting). The fundamental limitations of Stable Diffusion 1.5 in generating text could be the underlying cause of this constraint.

For ConceptFlow-M, our current method is not efficient enough regarding sampling speed and memory usage. Using a single NVIDIA A100 GPU, it takes an average of 9.3 seconds to generate an image of size 512x512. Moreover, the generation of multiple subjects solely based on a single prompt without additional guidance (*e.g.*, bounding boxes and local prompts) while ensuring no incorrect attribute binding still remains a challenge. This issue arises due to the potential problem of having similar semantic meanings between concepts in a prompt (Gu et al., 2024), as illustrated in Figure 25 when two human are mentioned. The face or outfit of a human is often binded incorrectly

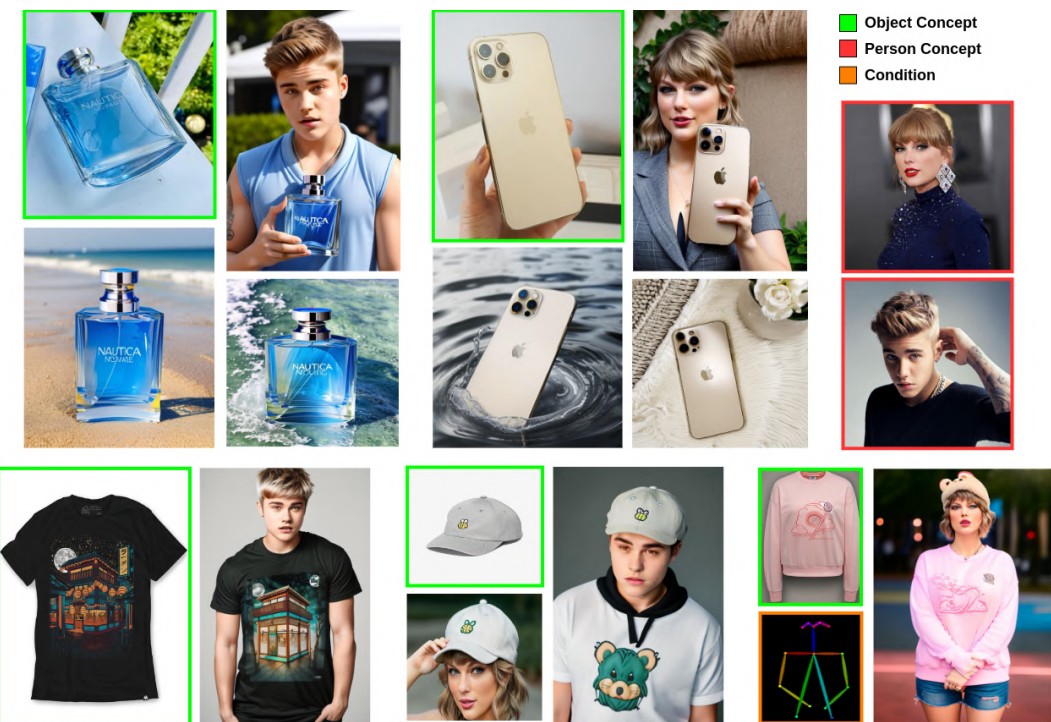

Figure 23: Illustration of our framework ConceptFlow for the applications on garment try-on and advertisement.

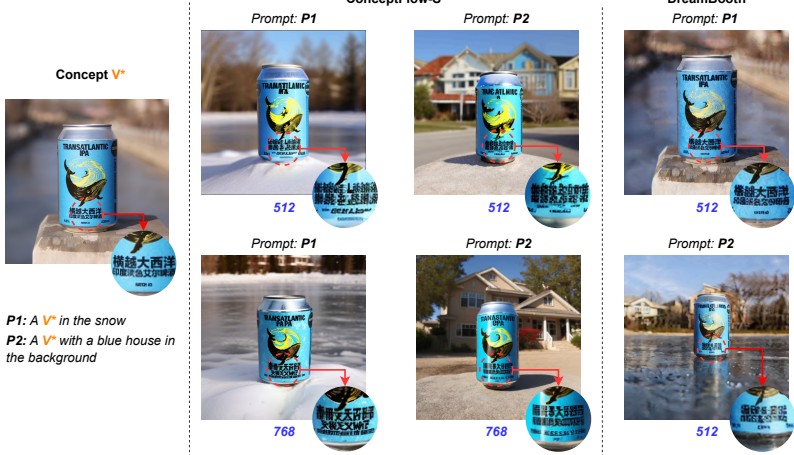

Figure 24: Limitation of ConceptFlow-S in capturing sotisphicated details of concepts such as text, logos in different training and generation resolution.

to the other human. Additionally, as the number of subjects mentioned in a prompt increases, resulting in a complex prompt (*e.g.*, "*A man wearing white glasses and wearing red shirt*"), the results as illustrated in Figure 26 tend to fail. This is due to the limitations of Stable Diffusion 1.5, which our framework is based on, in handling complex prompts. Currently, this limitation restricts our framework's ability to handle prompts with too many concepts.

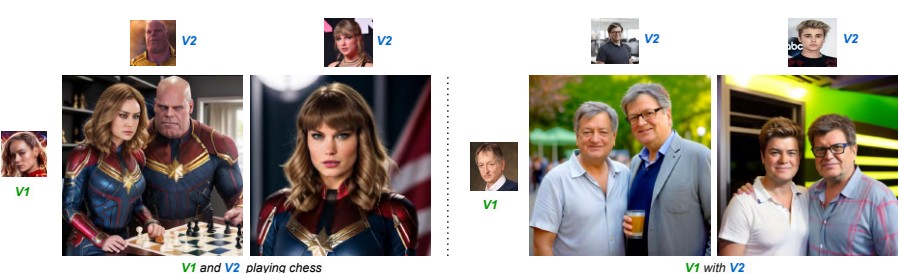

Figure 25: Limitation of ConceptFlow-M in multiple humans generation.

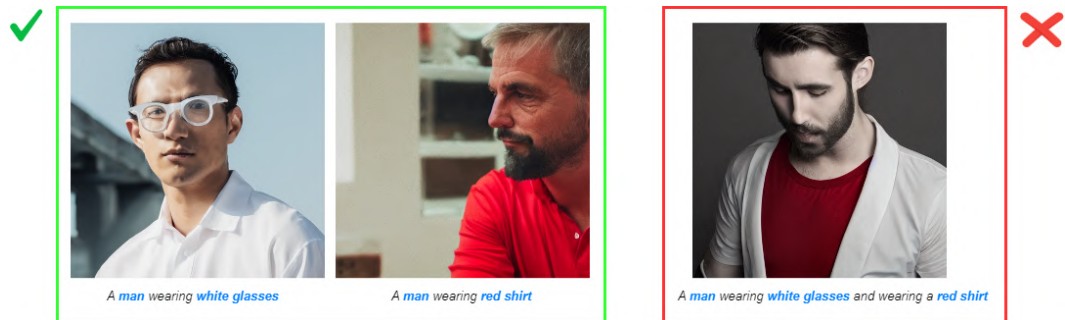

Figure 26: Limitation of Stable Diffusion 1.5 dealing with a complex prompt (more than 2 subjects).

