# OpenReview forum: "ConceptFlow: Unified Framework for Personalized Image Generation"
_ICLR.cc/2025/Conference — ICLR 2025 Conference Withdrawn Submission_

### Official Review · Reviewer_TYid · 2024-10-24

**Soundness:** 2
**Presentation:** 2
**Contribution:** 2
**Rating:** 5
**Confidence:** 5

**Summary:**

The paper proposes **ConceptFlow**, which includes two components: **ConceptFlow-S** for single-concept generation and **ConceptFlow-M** for multi-concept generation. ConceptFlow-S employs a **KronA-WED adapter** that integrates a Kronecker adapter with weight and embedding decomposition. It also introduces **attention regularization** to improve single-concept generation. ConceptFlow-M extends the models trained by ConceptFlow-S, using **SAMA** (Subject-Adaptive Matching Attention) and **layout consistency guidance** to handle multi-concept generation without requiring additional spatial conditions.

**Strengths:**

1. **Comprehensive personalization framework:**
  The paper covers both single and multi-concept personalization, supported by comparisons with recent works.

2. **Application insights and metrics:**
   The discussion includes practical applications, qualitative examples, and appropriate metrics for performance analysis.

**Weaknesses:**

1. **Over-complex Framework without Substantial Innovations:**
    - The **KronA-WED adapter** is like a simple stack of previous strategies, **Mix-of-Show’s *ED-LoRA*** [1] $+$ **Kronecker Adapter *(KronA)*** [6]  $\rightarrow$ **KronA-WED adapter**.
    - The proposed framework looks highly similar as a blend of **Mix-of-Show’s *gradient fusion*** [1] and **DisenBooth’s disentangled learning** [2], which dilutes its novelty.
    - Furthermore, the **attention regularization (AR)** employed seems highly similar as **Break-A-Scene's attention-based loss** [3], which is also designed to encourages that each handle attends only to the image region occupied by the corresponding concept.
    - The **SAMA module** for multi-concept generation largely reuses existing strategies, closely mirroring **DreamMatcher’s AMA** [4] (with **Eq. 6-9** resembling **Eq. 3, 4, 5, and 7** from DreamMatcher) but with minor modifications, such as adding a **foreground mask**.
    - The design of **Layout Guidance** follows the core idea of **A-Star's Attention Segregation Loss** [5], also not a novel technique.

    This accumulation of existing techniques makes the work feel like an industrial assembly, which reduces the perceived novelty of the contribution.

2. **Identity Preservation Issues and Low-Level Aesthetics:**
   - Several generated outputs display **identity dissimilarity** or **unsatisfactory aesthetics**, such as Justin Bieber, Emma Stone, cat in **Fig. 1**, which raise my concerns about the soundness of the proposed method and the robustness of the framework when dealing with **diverse or intricate concepts**.

3. **Dataset Limitations and Generalizability:**
   - The selection of subjects  in the manuscript are limited (**24 subjects (Fig. 12)**), which is limited and potentially biased toward **cherry-pick cases**. I believe these limited subjects can not represent the full spectrum of real-world complexities, and a broader test set covering **more diverse concepts and edge cases** would provide stronger evidence of the framework's generalizability.

**Reference**

[1] *Gu, et al. "Mix-of-show: Decentralized low-rank adaptation for multi-concept customization of diffusion models." NeurIPS (2024).*

[2] *Chen, et al. "Disenbooth: Identity-preserving disentangled tuning for subject-driven text-to-image generation." arXiv (2023).*

[3] *Avrahami, et al. "Break-a-scene: Extracting multiple concepts from a single image." SIGGRAPH 2023.*

[4] *Nam, et al. "Dreammatcher: Appearance matching self-attention for semantically-consistent text-to-image personalization." CVPR (2024).*

[5] *Agarwal, et al. "A-star: Test-time attention segregation and retention for text-to-image synthesis." ICCV (2023).*

[6] *Edalati, Ali, et al. "Krona: Parameter efficient tuning with kronecker adapter." arXiv (2022).*

**Questions:**

1. **Enhancing Identity Matching in Evaluation:**
   - The current evaluation focuses on **DINO scores** and **CLIP-T metrics**, but these might not capture **fine-grained identity preservation**, especially for faces. Have you considered incorporating face recognition tools like **FaceNet** or **ArcFace** [7]? These tools could offer bounding-box-level face similarity evaluation, providing a more precise measurement of face identity consistency.


2. **Expanding the Dataset and Subject Variety:**
   - As stated in  **Weakness 3**, The number of subjects used for both quantitative and qualitative evaluations appears limited (**Fig. 12**). This narrow selection may affect the statistical reliability of the results. Could you **experiment on more subjects or scenarios** to validate the performance of ConceptFlow across a broader range of real-world cases?

3. **Innovative Contributions of SAMA Compared to DreamMatcher:**
   - The *SAMA module* seems closely related to DreamMatcher’s *Appearance Matching Self-Attention (AMA)*, with the primary distinction being the use of a *concept foreground mask ($\mathbf{M}_k$)*. Could you further conduct **ablation studies on the AMA vs. SAMA** to show how it extends beyond DreamMatcher's approach?


4. **Time Efficiency and Computational Overhead:**
   - ConceptFlow involves multiple steps, such as *gradient fusion, SAMA matching, and layout consistency guidance*, which could introduce computational overhead. Could you provide a **memory / time efficiency comparison** between ConceptFlow and other state-of-the-art methods, such as **LoKr [8], DisenBooth [2], and Mix-of-Show [1]**? This would clarify the practical trade-offs between performance and efficiency for both single and multi-concept generation.


**Reference**

[7] *https://github.com/deepinsight/insightface*.

[8] *Yeh, et al. "Navigating text-to-image customization: From lycoris fine-tuning to model evaluation." ICLR (2023)*.

---

> ### Author Response · Authors · 2024-11-22
> **Response (1/2)**
>
> We thank the reviewer for the constructive comments and suggestions. We address your concerns point by point as follows.
>
> ### W1. The innovation of ConceptFlow
>
> For this point, please kindly see our global response above.
>
> ### W2. The identity dissimilarity or unsatisfactory aesthetics on generated outputs
>
> We agree with the reviewer that the identity dissimilarity or aesthetic quality of the generated outputs can vary, and one factor influencing this is the **generator seed**. In image generation, different seeds can result in distinct variations of the same prompt, and multiple attempts may yield better or more aesthetically pleasing results.
>
> ### W3, Q2. Expanding the Dataset and Subject Variety
>
> We thank the reviewer of the suggestion. Our dataset is heavily collected from the datasets of previous works on multi-concept generation [1,2,3] for human concepts, where they mainly use **a small set from 10-20 concepts**, and from DreamBench [4] for objects and animals, where we choose the concepts with intricate details for experiments.
>
> [1] Mix-of-Show: Decentralized Low-Rank Adaptation for Multi-Concept Customization of Diffusion Models, NeurIPS 2024
>
> [2] OMG: Occlusion-friendly Personalized Multi-concept Generation in Diffusion Models, ECCV 2024
>
> [3] FreeCustom: Tuning-Free Customized Image Generation for Multi-Concept Composition, CVPR 2024
>
> [4] DreamBooth: Fine Tuning Text-to-Image Diffusion Models for Subject-Driven Generation, CVPR 2023
>
> ### Q1. ArcFace metric for identity preservation evaluation on human concepts
>
> We thank the reviewer for the suggestion. We provide the ArcFace score for identity preservation evaluation on human concepts for both single and multiple concepts generation in the tables below:
>
> *Single concept generation:*
>
> | Methods          | ArcFace |
> |------------------|---------|
> | DreamBooth       | 0.305   |
> | Custom Diffusion | 0.172   |
> | DisenBooth       | 0.257   |
> | ED-LoRA          | 0.37    |
> | LoKr             | 0.375   |
> | **ConceptFlow-S**    | **0.397**   |
>
> *Multiple concepts generation:*
>
> | Method           | ArcFace |
> |------------------|---------|
> | Mix-Of-Show      | 0.223   |
> | Custom Diffusion | 0.169   |
> | FreeCustom       | 0.142   |
> | OMG              | 0.234   |
> | **ConceptFlow-M**    | **0.306**   |
>
> According to the results, ConceptFlow-S and ConceptFlow-M achieve a **higher ArcFace score** than the baseline methods, indicating that ConceptFlow can **better preserve the identity of human concepts**. This finding further demonstrates the effectiveness of our proposed **attention regularization (AR)** in ConceptFlow-S and **SAMA module** in ConceptFlow-M. Please kindly see **Section 3** and **Section 4** of our [**rebuttal manuscript**](https://anonymous.4open.science/r/ConceptFlow_ICLR25_Rebuttal-2411) for more analysis.
>
> ### Q3. Comparisons between SAMA and DreamMatcher's AMA in ConceptFlow-M pipeline
>
> We thank the reviewer for the suggestion. We showcase the qualitative results of this comparison in **Section 4** of our [**rebuttal manuscript**](https://anonymous.4open.science/r/ConceptFlow_ICLR25_Rebuttal-2411), and the quantitative results are as follow:
>
> | Method         | DINO | CLIP-T | ArcFace |
> |----------------|------|--------|---------|
> | ConceptFlow-M with AMA   | 0.442 | 0.728 | 0.236 |
> | **ConceptFlow-M with SAMA** | **0.454** | **0.784** | **0.306**
>
> The results show that **SAMA in ConceptFlow-M significantly outperforms AMA in DreamMatcher** in terms of both DINO and ArcFace scores, indicating that SAMA can **better preserve the identity of each concept**, especially for human concepts, in multi-concept images.

---

> ### Author Response · Authors · 2024-11-22
> **Response (2/2)**
>
> ### Q4. Time efficiency and computational overhead
>
> We provide the time efficiency and computational overhead of ConceptFlow in the Tables below:
>
> *Single concept learning:*
>
> We adopt an image adapter from DisenBooth and we store the attention maps of concept tokens for attention regularization (AR). The training time and the training memory of ConceptFlow-S are reported in the table below.
>
> | Method           | Training Time | Training Memory | Note |
> |------------------|---------------|-----------------|------|
> | DreamBooth       | 10 min        | 21 GB           | Fine-tune the whole SD U-Net |
> | Custom Diffusion | 6 min         | 16.5 GB         | Fine-tune K-V in cross-attn layers of U-Net |
> | DisenBooth       | 15 min        | 8.8 GB          | Fine-tune U-Net with LoRA ($r=4$), image adapter |
> | ED-LoRA          | 8 min         | 8.5 GB          | Fine-tune U-Net with LoRA ($r=4$), embedding decomposition (ED) |
> | LoKr             | 6 min         | 6 GB            | Fine-tune U-Net with KronA ($a_1=a_2=16$) |
> | **ConceptFlow-S**    | 15 min        | 10 GB           | Fine-tune U-Net with KronA ($a_1=a_2=16$), ED, image adapter, AR |
>
> *Multiple concepts generation:*
>
> ConceptFlow-M actually has to generate $(n + 1)$ images ($n$ reference images and 1 target image) at the same time for a prompt that involves $n$ concepts. We accomodate a loop for reference images generation to reduce the memory usage but it may increase the sampling time.
>
> | Method            | Fusion Time | Fusion Memory | Sampling Time | Sampling Memory |
> | ----------------- | ----------- | -------------|---------------|-----------------|
> | Mix-Of-Show     | 5 min        | 16.4 GB       | 2 s            | 6.4 GB          |
> | CustomDiffusion | 2 min        | 0             | 1.75 s         | 6.2 GB          |
> | FreeCustom      | 0           | 0             | 20 s           | 19 GB           |
> | OMG             | 0           | 0             | 15 s          | 13.5 GB         |
> | **ConceptFlow-M**   | 5 min        | 16.4 GB       | 9.3 s         | 25.7 GB         |

---

> ### Author Response · Authors · 2024-11-25
> **Looking forward to Reviewer's reply**
>
> Dear Reviewer,
>
> Thank you for your valuable feedback during the review process. Our rebuttal carefully addressed your concerns raised in the initial review. As the deadline for the discussion phase approaches, we kindly remind you to read our rebuttal. If you have any questions, suggestions, or further clarification on any points, please feel free to reach out.
>
> We look forward to your feedback and hope for a positive outcome.
>
> Thank you for your time and consideration.
>
> Best regards,
>
> Authors of Paper 4581

---

> > ### Comment · Reviewer_TYid · 2024-11-26
> > **Response to Authors' Rebuttal: Maintaining Current Score**
> >
> > The rebuttal addressed most of my questions and concerns, and I appreciate the clear explanation. I would like to maintain my current score.

---

> > > ### Author Response · Authors · 2024-11-27
> > >
> > > Dear Reviewer,
> > >
> > > We're pleased to address your concerns. Can you please consider update your rating?
> > >
> > > Thank you for your time and consideration.
> > >
> > > Best regards,
> > >
> > > Authors of Paper 4581

---

### Official Review · Reviewer_wm2m · 2024-11-03

**Soundness:** 2
**Presentation:** 2
**Contribution:** 1
**Rating:** 3
**Confidence:** 5

**Summary:**

This paper introduces ConceptFlow to balance identity preservation with prompt alignment, alleviating identity loss and concept omission.   The proposed KronA-WED adapter combined with a novel attention regularization objective balances reconstruction and editability
for single-concept generation. Additionally, the subject-adaptive matching attention module and layout consistency guidance strategy help prevent concept omission in multi-concept generation. Experimental results indicate that the proposed method has a positive impact on these issues to some extent.

**Strengths:**

1. This paper collects a small dataset.
2. The proposed method achieves good identity preservation.

**Weaknesses:**

1. Only a limited number of comparison methods, especially in multi-concept generation, does not adequately support the authors' claim. Why does this paper compare against so few published works? Let's leave aside arXiv papers, there are many accepted, peer-reviewed papers should be considered.

[1] FastComposer: Tuning-Free Multi-subject Image Generation with Localized Attention, IJCV

[2] FreeCustom: Tuning-Free Customized Image Generation for Multi-Concept Composition, CVPR 2024

[3] Key-Locked Rank One Editing for Text-to-Image Personalization, SIGGRAPH 2023

[4] Concept Weaver: Enabling Multi-Concept Fusion in Text-to-Image Models, CVPR2024

[5] OMG: Occlusion-friendly Personalized Multi-concept Generation in Diffusion Models, ECCV2024

[6] KOSMOS-G: Generating Images in Context with Multimodal Large Language Models, ICLR2024

2. Even in comparison with the methods listed in table, the proposed trivial method does not show significant improvement. This is particularly evident in CLIP-T, which evaluates editability and does not support the authors' claims of balancing the trade-off between reconstruction and editability. Can this paper provide metrics that evaluate this trade-off directly, rather than reporting separately when one aspect is better and the other is worse.

**Questions:**

1. It is good to see the new collected dataset. But for a comprehensive comprasion. Can the authors provide multi-concept generation results using CustomConcept101?
2. Most of papers report both CLIP-I and DINO. I believe use both metrics can better evaluate the identity preservation in different aspects.
3. I believe that extra conditions like boxes and masks can make generation more controllable. Of course, there are some situtations that condition-free generation is better. It would be appreciated to see the proposed can achieve better performance on both setting.

---

> ### Author Response · Authors · 2024-11-22
> **Response (1/2)**
>
> We thank the reviewer for the constructive comments and suggestions. We address your concerns point by point as follows.
>
> ### W1. Comparisons with other methods for multi-concept generation
>
> We thank the reviewer for the suggestion. We have included comparisons between ConceptFlow-M, FreeCustom [1], and OMG [2] for multi-concept generation in **Section 1** of our [**rebuttal manuscript**](https://anonymous.4open.science/r/ConceptFlow_ICLR25_Rebuttal-2411), as other methods either do not release their code or trained weights. Additionally, we use ArcFace metrics to evaluate identity preservation for human concepts.
>
> It is noteworthy that in a **condition-free setting**, ConceptFlow **outperforms** most other methods when generating multi-concept images.
> - For FreeCustom [1], this method requires reference images for concepts in the correct positions where it expects they to appear in the output image. As a result, using images with the concept centered, like those from a personalization dataset, leads to significant artifacts.
> - For OMG [2], a two-stage approach is used: first generating the layout and then integrating concept details into the layout. The main drawback of this approach occurs when the shape of the generated layout differs from the actual shape of the concept.
>
> [1] OMG: Occlusion-friendly Personalized Multi-concept Generation in Diffusion Models, ECCV 2024
>
> [2] FreeCustom: Tuning-Free Customized Image Generation for Multi-Concept Composition, CVPR 2024
>
> ### W2. Metrics that evaluate the trade-off directly
>
> We thank the reviewer for the suggestion.
> Since there is currently no work that directly addresses the metric evaluating the trade-off between reconstruction and editability, we compute the **F1 score between DINO and CLIP-T** as a provisional metric in the table below. The results show that **both ConceptFlow-S and ConceptFlow-M strike a better balance** between reconstruction (identity preservation) and editability (prompt alignment) compared to other methods.
>
> *Single concept generation:*
>
> | Method           | DINO  | CLIP-T | **F1-Score** |
> |------------------|-------|--------|----------|
> | DreamBooth       | **0.684** | 0.678  | 0.681    |
> | Custom Diffusion | 0.503 | **0.784**  | 0.613    |
> | DisenBooth       | 0.616 | *0.743*  | 0.674    |
> | ED-LoRA          | 0.667 | 0.703  | 0.685    |
> | LoKr             | 0.679 | 0.688  | 0.683    |
> | **ConceptFlow-S**    | *0.682* | 0.706  | **0.694**    |
>
> *Multiple concepts generation:*
>
> | Method           | DINO  | CLIP-T | **F1-Score** |
> |------------------|-------|-------|-------|
> | Mix-Of-Show      | 0.436 |0.779 | 0.559 |
> | Custom Diffusion  | 0.369 | **0.802** | 0.505 |
> | FreeCustom        | 0.369  | 0.722 | 0.480 |
> | OMG               | 0.357 | 0.732 | 0.480 |
> | **ConceptFlow-M**  | **0.454** | 0.784 | **0.575** |
>
> Additionally, we incorporate the ArcFace score to assess identity preservation for human concepts in **Section 1** of our [**rebuttal manuscript**](https://anonymous.4open.science/r/ConceptFlow_ICLR25_Rebuttal-2411). This metric also demonstrates the effectiveness in preversing human facial details of ConceptFlow-S and ConceptFlow-M.
>
> ### Q1. Multi-concept generation results on the CustomConcept101 dataset
>
> We thank the reviewer of the suggestion.
> Our dataset is heavily collected from the datasets of previous works on multi-concept generation [1,2,3] for human concepts, where they mainly use **a small dataset from 10-20 concepts**, and from DreamBench [4] for objects and animals, where we choose the concepts with intricate details for experiments.
> Therefore, conducting experiments on the CustomConcept101 dataset for ConceptFlow and other baseline methods for comparison would require substantial effort within this limited timeframe. However, we will take this suggestion into account for future work.
>
> [1] Mix-of-Show: Decentralized Low-Rank Adaptation for Multi-Concept Customization of Diffusion Models, NeurIPS 2024
>
> [2] OMG: Occlusion-friendly Personalized Multi-concept Generation in Diffusion Models, ECCV 2024
>
> [3] FreeCustom: Tuning-Free Customized Image Generation for Multi-Concept Composition, CVPR 2024
>
> [4] DreamBooth: Fine Tuning Text-to-Image Diffusion Models for Subject-Driven Generation, CVPR 2023

---

> ### Author Response · Authors · 2024-11-22
> **Response (2/2)**
>
> ### Q2. Report both CLIP-I and DINO for identity preservation evaluation
>
> We thank the reviewer for the suggestion. We showcase the results of CLIP-I and DINO for identity preservation evaluation in the tables below, where ConceptFlow-S and ConceptFlow-M both show **competitive results** compared to other methods:
>
> *Single concepts generation:*
>
> | Method           | CLIP-I | DINO  |
> |------------------|--------|-------|
> | DreamBooth       | *0.815*  | **0.684** |
> | Custom Diffusion | 0.701  | 0.503 |
> | DisenBooth       | 0.767  | 0.616 |
> | ED-LoRA          | 0.81  | 0.667 |
> | LoKr             | **0.826**  | 0.679 |
> | **ConceptFlow-S**    | 0.812  | *0.682* |
>
>
> *Multiple concepts generation:*
>
> | Method           | CLIP-I | DINO  |
> |------------------|--------|-------|
> | Mix-Of-Show      | 0.63   | 0.436 |
> | Custom Diffusion | 0.59   | 0.369 |
> | FreeCustom       | 0.574  | 0.36  |
> | OMG              | 0.593  | 0.357 |
> | **ConceptFlow-M**    | **0.637**  | **0.454** |
>
> However, as mentioned by Heiz at el.[1], the CLIP-I metric is not constructed to distinguish between different subjects that could have highly similar text descriptions (e.g. two different yellow clocks). Therefore, **it might not accurately reflect the identity preservation performance of the model**.
>
> [1] DreamBooth: Fine Tuning Text-to-Image Diffusion Models for Subject-Driven Generation, CVPR 2023
>
> ### Q3. About the extra conditions like boxes and masks
>
> We agree with the reviewer that additional conditions can enhance control over the generation process. However, in this work, we focus on the challenge of **condition-free generation** to provide greater flexibility and ease of use. While incorporating extra conditions into ConceptFlow is feasible, we plan to explore this in future research.

---

> ### Author Response · Authors · 2024-11-25
> **Looking forward to Reviewer's reply**
>
> Dear Reviewer,
>
> Thank you for your valuable feedback during the review process. Our rebuttal carefully addressed your concerns raised in the initial review. As the deadline for the discussion phase approaches, we kindly remind you to read our rebuttal. If you have any questions, suggestions, or further clarification on any points, please feel free to reach out.
>
> We look forward to your feedback and hope for a positive outcome.
>
> Thank you for your time and consideration.
>
> Best regards,
>
> Authors of Paper 4581

---

> > ### Comment · Reviewer_wm2m · 2024-11-25
> > **My concerns**
> >
> > Most of the concerns I raised remain unaddressed.
> >
> > W1. The comparison is still weak. KOSMOS-G, FastComposer, Key-Locked Rank One Editing for Text-to-Image Personalization are publicly available and have training scripts. I can find their github repository.
> >
> > W2. "There is currently no work that directly addresses the metric evaluating the trade-off between reconstruction and editability." As mentioned in the paper, the proposed method takes a good balance. However, I struggle to understand what this balance is (While you achieve a higher DINO score, the performance on CLIP-T is lower). It seems you do not outperform on both metrics simultaneously, and I don’t think a simple average score can adequately represent overall performance, as the metrics are fundamentally different in their measurements. Could you clarify this further? I believe introducing a new metric to evaluate the trade-off would be a significant contribution to this paper. A paper that can be accepted by a top-tier conference (i.e., ICLR) is not to say no existing works do something, but rather by providing convincing evidence for your claims (a good trade-off) with a higher DINO score and a lower CLIP-T score. By mentioning this, I don't mean you cannot use traditional metrics. However, you need to ensure the paper convincingly demonstrates how a higher DINO score and a lower score align with the claims made about achieving a balanced trade-off.
> >
> > Q1 and Q3 are left as future work, but I believe addressing them would enhance this paper significantly rather than deferring them to future work.

---

> ### Author Response · Authors · 2024-11-28
> **Response (1/2)**
>
> Dear Reviewer,
>
> We appreciate your fast and thoughtful reply. We would like to clarify some details as follows:
>
> ### W1. Comparisons with other methods for multi-concept generation
>
> For the methods you mentioned:
> - KOSMOS-G: The URLs provided for their trained weights are incorrect. Retraining their model would require significant computational resources (they used OpenImages V7 and InstructPix2Pix datasets).
> - FastComposer: The authors trained their model specifically for human concepts on the FFHQ dataset. Since we evaluate ConceptFlow on a broader range of concepts, including animals, objects, and humans, using their trained weights is not suitable. Additionally, training their model on a different dataset raises concerns about fairness.
> - Key-Locked Rank One Editing for Text-to-Image Personalization: We were only able to find an unofficial implementation of this method at [this repository](https://github.com/ChenDarYen/Key-Locked-Rank-One-Editing-for-Text-to-Image-Personalization). However, the codes there can not reproduce the multi-concept generation results presented in their paper.
> There are also several issues mentioned in that repository regarding this problem, such as [this issue](https://github.com/ChenDarYen/Key-Locked-Rank-One-Editing-for-Text-to-Image-Personalization/issues/9).
> As a result, using this implementation would not be a fair comparison.
>
> We also tried our best to compare our method with existing methods in multi-concept generation, including Custom Diffusion [1], FreeCustom [2], Mix-Of-Show [3], and OMG [4]. Please see the results in Table 1 in our
>  [**rebuttal PDF file**](https://anonymous.4open.science/r/ConceptFlow_ICLR25_Rebuttal-2411). Our method achieved better performance than these state-of-the-art methods.
>
> [1] Multi-Concept Customization of Text-to-Image Diffusion, CVPR 2023
>
> [2] FreeCustom: Tuning-Free Customized Image Generation for Multi-Concept Composition, CVPR 2024
>
> [3] Mix-of-Show: Decentralized Low-Rank Adaptation for Multi-Concept Customization of Diffusion Models, NeurIPS 2024
>
> [4] OMG: Occlusion-friendly Personalized Multi-concept Generation in Diffusion Models, ECCV 2024
>
>
> ### W2. What "balance trade-off" means
>
> Fine-tuning-based methods often face challenges in balancing the **trade-off** between reconstruction and editability, i.e. **achieving a higher DINO score can result in a decrease in the CLIP-T score, and vice versa**. Additionally, qualitative results (Figure 1 in our [**rebuttal manuscript**](https://anonymous.4open.science/r/ConceptFlow_ICLR25_Rebuttal-2411/ConceptFlow_Rebuttal_ICLR25.pdf) underscore this trade-off: DreamBooth achieves strong reconstruction fidelity but poorly aligns with prompts, while Custom Diffusion aligns well with prompts but exhibits weak reconstruction quality.
>
> Quantitatively, Table 1 of our rebuttal manuscript provides further evidence. Comparing with ED-LoRA [1], DreamBooth [2] achieves a higher DINO score but suffers from a lower CLIP-T score. A similar trend is observed across other methods in this table.
>
> In contrast, ConceptFlow-S achieves both higher DINO and CLIP-T scores compared to parameter-efficient fine-tuning methods (ED-LoRA [1] and LoKR [3]), demonstrating that **ConceptFlow-S can simultaneously enhance these two capabilities**.
>
> While ConceptFlow-S has a lower CLIP-T score compared to Custom Diffusion [4] and DisenBooth [5], **it significantly outperforms these methods in DINO score**. These methods overlooked the importance of reconstruction capability. Additionally, when compared to DreamBooth [2], ConceptFlow-S showcases a **competitive DINO score along with a better CLIP-T score**. DreamBooth [2] is often affected by overfitting, which limits its ability to effectively balance the trade-off.
>
> Furthermore, as the suggestion by Reviewer wm2m, we compute **the F1 score between DINO and CLIP-T as a provisional metric directly evaluating the trade-off** between reconstruction and editability. Our method outperforms existing methods in terms of F1 score, highlighting the superior balance of our method. Please see the results in Table 1 in our
>  [**rebuttal PDF file**](https://anonymous.4open.science/r/ConceptFlow_ICLR25_Rebuttal-2411).
>
>
> [1] Mix-of-Show: Decentralized Low-Rank Adaptation for Multi-Concept Customization of Diffusion Models, NeurIPS 2024
>
> [2] DreamBooth: Fine Tuning Text-to-Image Diffusion Models for Subject-Driven Generation, CVPR 2023
>
> [3] Navigating Text-To-Image Customization: From LyCORIS Fine-Tuning to Model Evaluation, ICLR 2024
>
> [4] Multi-Concept Customization of Text-to-Image Diffusion, CVPR 2023
>
> [5] DisenBooth: Identity-Preserving Disentangled Tuning for Subject-Driven Text-to-Image Generation, ICLR 2024

---

> > ### Author Response · Authors · 2024-11-28
> > **Response (2/2)**
> >
> > ### Q1 and Q3:
> >
> > Regarding **conducting experiments on the CustomConcept101 dataset**, this is a major request. ***This dataset is too large to conduct experiments within this limited timeframe of the rebuttal process***. In addition, our work mainly handle the interaction between the human and object while this dataset has a few human concepts. However, we can add the experiments in the camera ready.
> >
> > Regarding **adding extra conditions**, this is a major request. Our work tackles the issues of ***condition-free generation*** to reduce the constraints for users. We aim to provide greater flexibility and ease of use than works needing conditions like boxes and masks. ***Integrating extra conditions is out-of-the-scope of this work***. Adding conditions is not simply plug-and-play modules. It require to modify and optimize some parts in the framework to achieve good performance, which takes time and not suitficient for limited timeframe of the rebuttal process. Therefore, we'll leave this part for the future work.
> >
> > We hope these clarifications resolve your concerns and are happy to address further questions.
> >
> > Best regards,
> >
> > Authors of Paper 4581

---

> > > ### Comment · Reviewer_wm2m · 2024-11-28
> > >
> > > Thank you for your rebuttal.
> > >
> > > "Integrating extra conditions is out-of-the-scope of this work." which means that your work is difficult to achieve a more flexible image generation (One may want to control the region of generation sometimes, and not other times). While there are so many works enabling both conditional and condition-free generation, this work is specifically limited to condition-free generation.
> > >
> > > In the paper, you state "When it comes to multiple concepts, creating images from a single prompt without extra conditions, such as layout boxes or semantic masks, is problematic due to significantly identity loss and concept omission." While this is an important challenge to address, it would be ideal to do so without sacrificing the condition-based generation ability. For a top-tier conference like ICLR, I would like to see submissions that can address the question the submission raised without compromising its existing strengths. It would be a stronger contribution than tackling one issue at the expense of another.
> > >
> > > Bests,
> > > Reviewer wm2m

---

### Official Review · Reviewer_jVBs · 2024-11-03

**Soundness:** 3
**Presentation:** 3
**Contribution:** 3
**Rating:** 5
**Confidence:** 4

**Summary:**

The manuscript introduces propose two strategies "ConceptFlow-S" and "ConceptFlow-M" for single-concept generation and multiple-concept generation to resolve the limitation of identity preservation, prompt alignment, and concept omission from previous methods. For single concept, the authors introduce KronA-WED adapter with the attention regularization objective. For multiple concept, the authors propose the SAMA module and layout the consistency guidance. This method facilitates rapid and efficient single and multi-subject customization. The results demonstrate that the method is competitive with leading frameworks in various image generation tasks.

**Strengths:**

1. The work identifies the issues in both single and multiple concept personalization such as trade off between reconstruction and edibility, and identity loss and the issue of concept missing.

2. The authors have developed a framework for generating personalized images that effectively integrates strategies to solve both single (ConceptFlow-S) and multiple concept (ConceptFlow-M).

2. The paper provides experimental results, including both quantitative and qualitative assessments, showcasing the superior performance of the framework. The results clearly highlight the effectiveness of the proposed method in facilitating personalized image generation.

**Weaknesses:**

1. In Figure 5, the method utilizes SAMA module to enhance the identity details. However, it is unclear how it would perform with fine-grained subjects (two dogs or two cats with different breed). Also, I wonder how this module would work when the concept size increases. Clarification is needed on whether the module can effectively manage such fine distinctions and multiple diverse subjects.

2. Recent methodologies [1, 2] have demonstrated the capability to learn multi-concept personalization, it remains uncertain if the proposed work can handle multiple personalized instances (> 2), particularly for contexts involving up to five subjects. Although quantitative results for two subjects are provided in both paper and appendix, the absence of qualitative results for three or more subjects in both the main text and appendix might be a notable omission. Including these results would substantiate the method's capability in more complex scenarios.


[1] Liu, Zhiheng, et al. "Cones 2: Customizable image synthesis with multiple subjects." arXiv preprint arXiv:2305.19327 (2023).

[2] Yeh, Chun-Hsiao, et al. "Gen4Gen: Generative Data Pipeline for Generative Multi-Concept Composition." arXiv preprint arXiv:2402.15504 (2024).

**Questions:**

Given the concerns mentioned, particularly around the method's scalability to more complex multi-subject personalizations and the clarification behind module choices, I recommend a "marginally below the acceptance threshold" for this paper. Enhancements in demonstrating multi-subject capabilities, clarity in embedding visualization, and justification for the choice of technology could potentially elevate the manuscript to meet publication standards.

---

> ### Author Response · Authors · 2024-11-22
> **Response (1/1)**
>
> We thank the reviewer for the constructive comments and suggestions. We address your concerns point by point as follows.
>
> ### W1. How SAMA performs with fine-grained subjects (two dogs or two cats with different breeds)
>
> Firstly, the effectiveness of SAMA mainly relies on the accuracy of concept masks, i.e. **the cross-attention maps of their tokens**. We consider the combinations such as two dogs, two cats, dog and cat, two humans, as **semantically semantic concepts**.  In these cases, Stable Diffusion 1.5 (SD 1.5) often suffers from **attribute binding problem** [1], where the attention map of one token is incorrectly focused on the other token even at the begining of the denoising process. Therefore, SAMA, and also ConceptFlow-M, tends to fail in these cases. We carefully mention this in **Section 5.1** of our [**rebuttal manuscript**](https://anonymous.4open.science/r/ConceptFlow_ICLR25_Rebuttal-2411).
>
> Nonetheless, for the case of **concepts with intricate details**, SAMA demonstrates it capability of preserving complex patterns, as shown in **Section 3** of our [**rebuttal manuscript**](https://anonymous.4open.science/r/ConceptFlow_ICLR25_Rebuttal-2411).
>
> [1] Training-Free Structured Diffusion Guidance for Compositional Text-to-Image Synthesis, ICLR 2023
>
> ### W1. How SAMA works when the concept size increases
>
> In theory, our proposed SAMA module **can work for a general case of $N$ concepts**. Specifically:
> - For each concept $k$ ($1 \leq k \leq N$):
>     - We have a process to generate one image of it (which is called *reference image*). The reference features corresponding to this concept at each timestep, $\psi^{ref}_k$, are calculated from the intermediate features within U-Net of this process.
>     - Its foreground mask $\mathbf{M}_k$ is extracted from the cross attention maps of its tokens in the target denoising process.
>     - The warping calculation to obtain $V_{k}^{ref \to trg}$ is calculated follow the equation (6), (7) in our manuscript.
> - From all warped values $V_{k}^{ref \to trg} (k=1,2,..,N)$, we perform the blending operation using concept masks $\{\mathbf{M}_k\}_{k=1}^N$ to obtain the final value for the self-attention calculation in the target denoising process, following equation (8) in the manuscript.
>
> ### W2. Results on larger number than two concepts
>
> In scenarios involving more than two concepts, ConceptFlow-M produces both successful and failed outputs with roughly equal probability. It is noteworthy that **most condition-free multi-concept generation methods [1,2] on Stable Diffusion 1.5 (SD 1.5) exhibit the same issue**, and they only conduct experiments with two concepts.
> We are convinced that this limitation stems from SD 1.5's challenges in handling complex prompts (i.e., prompts involving more than two concepts). For visualization and further analysis, please refer to **Section 5.2** in our [**rebuttal manuscript**](https://anonymous.4open.science/r/ConceptFlow_ICLR25_Rebuttal-2411).
>
> To address this problem, typical approaches involve incorporating conditions such as boxes, masks, and applying local prompting and denoising [3,4]. However, in this work, *our focus for ConceptFlow is on advancing **condition-free generation***. As a result, addressing this limitation will be a potential area for future work.
>
> [1] Multi-Concept Customization of Text-to-Image Diffusion, CVPR 2023
>
> [2] Key-Locked Rank One Editing for Text-to-Image Personalization, SIGGRAPH 2023
>
> [3] Mix-of-Show: Decentralized Low-Rank Adaptation for Multi-Concept Customization of Diffusion Models, NeurIPS 2024
>
> [4] OMG: Occlusion-friendly Personalized Multi-concept Generation in Diffusion Models, ECCV 2024

---

> > ### Comment · Reviewer_jVBs · 2024-11-27
> > **Response to Author's Rebuttal**
> >
> > Thanks for the authors' response. I have no further comments regarding the SAMA-related questions. However, I still have concerns about the multiple concept generation aspect. While the authors mention that ConceptFlow focuses on advancing condition-free generation and that addressing this limitation is a potential direction for future work, I believe this issue is closely related to the current work and should not be deferred entirely to future exploration.
> >
> > Have the author's tried to generate 3 or 4 different concepts (condition-free)? I am still curious that the method could have controllability to handle the generation beyond 2 concepts (mostly shown in the paper).

---

> > > ### Author Response · Authors · 2024-11-28
> > >
> > > Dear Reviewer,
> > >
> > > We are happy when your concerns were addressed.
> > >
> > > Regarding generating 3 or 4 concepts, our work mainly tackles generating two concepts, which is the norm in multiple concept generation [1][2]. When generating more than two concepts, the results of our framework tends to unsatisfied. We found that this is due to the limitations of Stable Diffusion 1.5, which our framework is based on, in handling complex prompts. Currently, this limitation restricts our framework’s ability to handle prompts with too many concepts.
> > >
> > > Please kindly see **Section 5.2** in our [**rebuttal PDF file**](https://anonymous.4open.science/r/ConceptFlow_ICLR25_Rebuttal-2411) for more analysis and illustration figures.
> > >
> > > [1] Multi-Concept Customization of Text-to-Image Diffusion, CVPR 2023
> > >
> > > [2] OMG: Occlusion-friendly Personalized Multi-concept Generation in Diffusion Models, ECCV 2024
> > >
> > > Best regards,
> > >
> > > Authors of Paper 4581

---

> ### Author Response · Authors · 2024-11-25
> **Looking forward to Reviewer's reply**
>
> Dear Reviewer,
>
> Thank you for your valuable feedback during the review process. Our rebuttal carefully addressed your concerns raised in the initial review. As the deadline for the discussion phase approaches, we kindly remind you to read our rebuttal. If you have any questions, suggestions, or further clarification on any points, please feel free to reach out.
>
> We look forward to your feedback and hope for a positive outcome.
>
> Thank you for your time and consideration.
>
> Best regards,
>
> Authors of Paper 4581

---

### Official Review · Reviewer_2SBt · 2024-11-04

**Soundness:** 2
**Presentation:** 2
**Contribution:** 1
**Rating:** 3
**Confidence:** 5

**Summary:**

This paper introduces the ConceptFlow framework, including the ConceptFlow-S component for robust single-concept learning and generation, and the ConceptFlow-M component to create images of multiple concepts without the need for spatial guidance.

**Strengths:**

ConceptFlow-M needs no spatial guidance and can somehow handle the missing concept issue, which is important in multi-concept personalized image generation.

**Weaknesses:**

1. This work claims to be a unified framework of personalized image generation in the title, but there is no presentation about how it **unifies** personalized image generation.
2. The contribution of the proposed method is weak. There are too many modules that already exist with no relation to each other.
3. The comparison experiments are not rational. Since the proposed model introduces an image adapter, it should compare with the methods using the image adapter. In fact, as a fine-tune-based method, there is no need to use an image adapter since it needs an image input during the inference. I suggest the authors compare the proposed method with some zero-shot methods like IP-Adapter [1], SSR-Encoder [2], and MS-Diffusion [3], which can also solve the same issues.
4. The experiments are completely not enough with few baselines and qualitative examples.

[1] Ye, Hu, et al. "Ip-adapter: Text compatible image prompt adapter for text-to-image diffusion models." arXiv preprint arXiv:2308.06721 (2023).

[2] Zhang, Yuxuan, et al. "Ssr-encoder: Encoding selective subject representation for subject-driven generation." Proceedings of the IEEE/CVF Conference on Computer Vision and Pattern Recognition. 2024.

[3] Wang, X., et al. "MS-Diffusion: Multi-subject Zero-shot Image Personalization with Layout Guidance." arXiv preprint arXiv:2406.07209 (2024).

**Questions:**

Please see the weakness.

---

> ### Author Response · Authors · 2024-11-22
> **Response (1/1)**
>
> We thank the reviewer for the constructive comments and suggestions. We address your concerns point by point as follows.
>
> ### W1, W2. How ConceptFlow unifies personalized image generation. The contribution of the proposed method and the relation of modules.
>
> For these two points, please kindly see our global response above.
>
> ### W3. Compare with methods that also use image adapter
>
> Firstly, in ConceptFlow-S, we adopt the image adapter and related loss objectives from DisenBooth [1] **during fine-tuning process** for better disentangled learning of identity-relevant andi identity-irrelevant information. **We do not use the image adapter for inference** like other zero-shot methods [2,3,4].
>
> We present a qualitative comparison between ConceptFlow-S and zero-shot methods [2,3] in **Section 2** of our [**rebuttal manuscript**](https://anonymous.4open.science/r/ConceptFlow_ICLR25_Rebuttal-2411). Since MS-Diffusion [4] does not rely on Stable Diffusion 1.5, we do not include them in the comparison. The results show that **ConceptFlow-S generates images with superior identity preservation for concepts with intricate details**, which is a challenge commonly faced by zero-shot methods. We remark that Ip-adapter [3] and MS-Diffusion [4] have not been published in peer-review venues.
>
> [1] DisenBooth: Identity-Preserving Disentangled Tuning for Subject-Driven Text-to-Image Generation, ICLR 2024
>
> [2] Ssr-encoder: Encoding selective subject representation for subject-driven generation, CVPR 2024
>
> [3] Ip-adapter: Text compatible image prompt adapter for text-to-image diffusion models, arXiv preprint arXiv:2308.06721 (2023).
>
> [4] MS-Diffusion: Multi-subject Zero-shot Image Personalization with Layout Guidance, arXiv preprint arXiv:2406.07209 (2024).
>
> ### W4. More experiments results
>
> We thank the reviewer for the suggestion. Please see the **Section 1** in our [**rebuttal manuscript**](https://anonymous.4open.science/r/ConceptFlow_ICLR25_Rebuttal-2411), where we add additional experiments results, especially for ConceptFlow-M, to demonstrate the effectiveness of the ConceptFlow framework.

---

> ### Author Response · Authors · 2024-11-25
> **Looking forward to Reviewer's reply**
>
> Dear Reviewer,
>
> Thank you for your valuable feedback during the review process. Our rebuttal carefully addressed your concerns raised in the initial review. As the deadline for the discussion phase approaches, we kindly remind you to read our rebuttal. If you have any questions, suggestions, or further clarification on any points, please feel free to reach out.
>
> We look forward to your feedback and hope for a positive outcome.
>
> Thank you for your time and consideration.
>
> Best regards,
>
> Authors of Paper 4581

---

### Author Response · Authors · 2024-11-22
**Global Response (1/2)**

We thank the reviewers for their constructive comments and suggestions. We are encouraged with positive feedback that ConceptFlow **effectively solves both single and multiple concept generation problems** (jVBs, 2SBt, TYid), has **good identity preservation capability** (wm2m), and has **potential applications** (TYid).

In our [**rebuttal manuscript**](https://anonymous.4open.science/r/ConceptFlow_ICLR25_Rebuttal-2411) (clickable), we have added additional experiments, ablation studies, analyses, and discussions based on the suggestions from all reviewers:
1. Additional experiments
2. ConceptFlow-S versus zero-shot methods in single concept generation
3. Attention Regularization (AR) in ConceptFlow-S versus Break-A-Scene's strategy
4. Subject Adaptive Matching Attention (SAMA) in ConceptFlow-M versus DreamMatcher's AMA
5. Limitation of ConceptFlow-M:
    - Generations of multiple semantically similar concepts
    - Generations of more than two concepts

Also, we would like to highlight the **contributions** of ConceptFlow, as well as how it **unifies personalized image generation**.

## A comprehensive framework for personalized image generation

Firstly, ConceptFlow-S finetunes Stable Diffusion for **robust single concept learning and generation**. We can then combine the individual weights learned by ConceptFlow-S, enabling the generation of multi-concept images through the ConceptFlow-M **without requiring of extra conditions** (e.g. boxes, masks).

It is noteworthy that **using weights learned by ConceptFlow-S** significantly enhances the performance of ConceptFlow-M in generating multiple concepts compared to other single-concept learning methods. Additionally, each concept only needs to be finetuned once, with the trained weights saved for future multi-concept generation. Therefore, **ConceptFlow unifies personalized image generation**, seamlessly eveloving from single to multi-concept generation.

With superiority in generating interactions between **human and object**, as well as between **human and animal**, ConceptFlow showcases its **potential applications** in areas such as advertising, storytelling, and even garment synthesis.

---

> ### Author Response · Authors · 2024-11-22
> **Global Response (2/2)**
>
> ## Core contributions and novelty
>
> ### 1. Develop a novel yet robust single-concept learning pipeline.
>
> We introduce **KronA-WED** adapter, which leverages KronA [1], DORA [2] and embedding decomposition [3]:
> - KronA [1] relaxes the low-rank assumption in LoRA [4], improving the learning capability of parameter-efficient fine-tuning (PEFT) process while maintaining a smaller model size than LoRA.
> - DORA [2] further enhances the performance gains from PEFT methods via the weight decomposition technique.
> - Embedding decomposition [3] has an important role for the success of Gradient Fusion [5], which we use to combine the individual weights learned by ConceptFlow-S. This merged weight is then utilized at the beginning of ConceptFlow-M.
>
> We leverage Disentangled learning [6] to prevent the model from learning concept-irrelevant details, such as background scenes, thus enhancing editability. We employ an image adapter with three loss objectives **only during the finetuning process**, but not during inference like existing methods [7, 8, 9].
>
> [1] KronA: Parameter Efficient Tuning with Kronecker Adapter, NeurIPS 2023 Workshop
>
> [2] DoRA: Weight-Decomposed Low-Rank Adaptation, ICML 2024
>
> [3] P+: Extended Textual Conditioning in Text-to-Image Generation, arXiv 2023
>
> [4] LoRA: Low-Rank Adaptation of Large Language Models, ICLR 2022
>
> [5] Mix-of-Show: Decentralized Low-Rank Adaptation for Multi-Concept Customization of Diffusion Models, NeurIPS 2024
>
> [6] DisenBooth: Identity-Preserving Disentangled Tuning for Subject-Driven Text-to-Image Generation, ICLR 2024
>
> [7] Ip-adapter: Text compatible image prompt adapter for text-to-image diffusion models, arXiv preprint arXiv:2308.06721 (2023).
>
> [8] Ssr-encoder: Encoding selective subject representation for subject-driven generation. CVPR 2024
>
> [9] MS-Diffusion: Multi-subject Zero-shot Image Personalization with Layout Guidance, arXiv preprint arXiv:2406.07209 (2024).
>
> ### 2. Novel attention regularization (AR) in ConceptFlow-S
>
> Attention regularization during fine-tuning Stable Diffusion is a **widely used technique** for adjusting attention maps of specific tokens in various tasks, such as subject-driven personalization [1,2], relation inversion [3], and concept erasure [4].
>
> In ConceptFlow-S, we propose a novel, part-of-speech-inspired method to handle the attention maps of the adjective token ($V_{rand}$) and the noun token ($V_{class}$) of a concept differently. Specifically:
> - The noun token $V_{class}$ should align with the extracted masks from training input images
> - The adjective token $V_{rand}$ can activate **some specific regions** within those masks.
>
> This approach is **particularly effective for learning human concepts**, where the "identity" of the desired concept can vary in each training image (e.g., different clothing or hairstyles). The relaxed constraints on the adjective token allow the model to better capture the common features of the concept, which, in the case of human concepts, is primarily the face (please see **Section 3** in our [**rebuttal manuscript**](https://anonymous.4open.science/r/ConceptFlow_ICLR25_Rebuttal-2411) for experimental results).
>
> [1] Break-A-Scene: Extracting Multiple Concepts from a Single Image, SIGGRAPH Asia 2023
>
> [2] FastComposer: Tuning-Free Multi-Subject Image Generation with Localized Attention, IJCV 2024
>
> [3] Customizing Text-to-Image Generation with Inverted Interaction, ACM MM’24
>
> [4] MACE: Mass Concept Erasure in Diffusion Models, CVPR 2024
>
> ### 3. Subject Adaptive Matching Attention (SAMA) in ConceptFlow-M
>
> Inspired by the AMA module in DreamMatcher [1], we extend this strategy to multi-concept generation in ConceptFlow-M to **address the identity loss problem** for concepts in multi-concept images.
>
> The modifications from AMA involve using concept attention maps as the "foreground mask" to calculate the displacement fields in each warping operation, particularly in the **masking step during the calculation of matching cost volume $\mathbf{C}_k$**. This enhances the accuracy of the feature similarity calculation between the target image and each reference image (please see **Section 4** in our [**rebuttal manuscript**](https://anonymous.4open.science/r/ConceptFlow_ICLR25_Rebuttal-2411) for experimental results).
>
> [1] DreamMatcher: Appearance Matching Self-Attention for Semantically-Consistent Text-to-Image Personalization, CVPR 2024
>
> ### 4. Layout consistency guidance in ConceptFlow-M
>
> From the investigation that the **concept missing issue** in personalized multi-concept generation is primarily due to the model's difficulty in retaining the layout from initial to final denoising steps, inspired by [1], we develop a simple yet effective **layout consistency guidance** mechanism in ConceptFlow-M.
>
> [1] A-STAR: Test-time Attention Segregation and Retention for Text-to-image Synthesis, ICCV 2023.

---

### Note · Authors · 2025-06-01

I have read and agree with the venue's withdrawal policy on behalf of myself and my co-authors.

---

### Meta-Review · Area_Chair_uCnh · 2024-12-19

**Metareview:**

- **Strengths**:
  - Comprehensive framework covering both single and multi-concept personalization.
  - Demonstrated improvements in identity preservation and editability with robust quantitative metrics.

- **Weaknesses**:
  - Limited novelty in methodology.
  - Scalability concerns and dataset constraints impact generalizability.
  - Incremental improvements over prior art, with some claims inadequately substantiated.

Despite addressing most issues during the rebuttal, the concerns about scalability and generalizability weighed heavily in the final decision. The work was evaluated as promising but needing further development for inclusion in a top-tier venue.

**Additional Comments On Reviewer Discussion:**

## Reviewer Feedback and Points Raised

1. **Weak Comparisons** (Reviewer 2SBt, wm2m):
   - Insufficient baseline comparisons, especially for multi-concept generation.
   - Limited methods evaluated (only a few recent works) and lacking certain state-of-the-art methods (e.g., KOSMOS-G).

2. **Metrics and Trade-off Evaluation** (Reviewer wm2m, TYid):
   - The paper claims to balance identity preservation and prompt alignment but fails to demonstrate this with a unified trade-off metric.

3. **Over-complex Framework with Limited Novelty** (Reviewer TYid):
   - Techniques like KronA-WED, SAMA, and Layout Consistency were deemed incremental and heavily inspired by prior works (e.g., DreamMatcher, Break-A-Scene).

4. **Scalability Issues** (Reviewer jVBs):
   - Concerns about the method's scalability beyond two concepts.
   - Limited qualitative results for more complex scenarios involving multiple concepts.

5. **Dataset Limitations** (Reviewer TYid):
   - Narrow subject variety potentially led to cherry-picking. A more diverse dataset was requested.

6. **Identity Preservation Issues** (Reviewer TYid):
   - Identity dissimilarity in certain outputs (e.g., human faces) raised concerns about robustness.

7. **Computational Overhead** (Reviewer TYid):
   - Questions about the time and memory efficiency of the proposed framework compared to others.

## Author Responses and Revisions

1. **Additional Experiments and Baselines**:
   - Added comparisons with methods like FreeCustom and OMG, with improved results showcased using metrics like ArcFace.
   - Clarified reasons for excluding some methods (e.g., KOSMOS-G due to lack of reproducibility or computational constraints).

2. **Trade-off Demonstrations**:
   - Introduced an F1 score to balance identity preservation (DINO) and editability (CLIP-T), demonstrating improvements over other methods.

3. **Technical Clarifications**:
   - Highlighted how SAMA differs from DreamMatcher’s AMA, focusing on the inclusion of a concept foreground mask, which showed improved performance.

4. **Scalability and Limitations**:
   - Acknowledged limitations in handling more than two concepts and proposed future work to address scalability challenges.

5. **Dataset and Subject Variety**:
   - Acknowledged limitations in dataset diversity and committed to broader evaluations in future work.

6. **Efficiency Analysis**:
   - Provided computational benchmarks, demonstrating competitive efficiency compared to baseline methods like DisenBooth and DreamBooth.

---

### Decision · Program_Chairs · 2025-01-22

Reject